# *MMRole*: A Comprehensive Framework for Developing and Evaluating Multimodal Role-Playing Agents

**Yanqi Dai**[1], **Huanran Hu**[2], **Lei Wang**[1], **Shengjie Jin**[1], **Xu Chen**[1]*, **Zhiwu Lu**[1]*
[1]Gaoling School of Artificial Intelligence, Renmin University of China
[2]College of Information and Electrical Engineering, China Agricultural University

## Abstract

Recently, Role-Playing Agents (RPAs) have garnered increasing attention for their potential to deliver emotional value and facilitate sociological research. However, existing studies are primarily confined to the textual modality, unable to simulate humans' multimodal perceptual capabilities. To bridge this gap, we introduce the concept of Multimodal Role-Playing Agents (MRPAs), and propose a comprehensive framework, *MMRole*, for their development and evaluation, which comprises a personalized multimodal dataset and a robust evaluation approach. Specifically, we construct a large-scale, high-quality dataset, *MMRole-Data*, consisting of 85 characters, 11K images, and 14K single or multi-turn dialogues. Additionally, we present a robust evaluation approach, *MMRole-Eval*, encompassing eight metrics across three dimensions, where a reward model is designed to score MRPAs with the constructed ground-truth data for comparison. Moreover, we develop the first specialized MRPA, *MMRole-Agent*. Extensive evaluation results demonstrate the improved performance of *MMRole-Agent* and highlight the primary challenges in developing MRPAs, emphasizing the need for enhanced multimodal understanding and role-playing consistency. The data, code, and models are all available.[1]

## 1 Introduction

The advancement of large language models (LLMs) (Zhao et al., 2023) has significantly catalyzed the rise of Role-Playing Agents (RPAs) (Chen et al., 2024b), which are engineered to emulate specific characters and engage in dialogues with human users or other characters. Unlike AI productivity assistants, RPAs primarily focus on delivering emotional value (Li et al., 2023a; Wang et al., 2023; Shao et al., 2023) and facilitating sociological research (Zhou et al., 2023b; Wang et al., 2024c; Chen et al., 2024a; Gu et al., 2024), where typical applications include emotional companions, NPCs in video games, digital clones, and social simulations.

The primary characteristic of RPAs is their capability to engage in human-like and immersive interactions. However, existing studies in role-playing are primarily confined to the textual modality, which has considerable limitations. In the real-world context, human perception integrates multiple modalities, especially visual and textual, allowing for a more direct and comprehensive understanding of the environment than text alone could provide. Therefore, enhancing RPAs with multimodal capabilities is a crucial next step for conducting more realistic and engaging interactions.

In this paper, we introduce the concept of Multimodal Role-Playing Agents (MRPAs). MRPAs are designed to emulate specific characters and engage in dialogues centered around images, with either human users or other characters. Furthermore, we propose *MMRole*, a comprehensive framework for developing and evaluating MRPAs. As presented in Figure 1, this framework includes two principal components: a large-scale, high-quality dataset and a robust evaluation approach for MRPAs.

**Dataset Construction:** The dataset for MRPAs comprises character profiles, images, and dialogues centered around images. Specifically, we consider three categories of characters: fictional characters, historical and public figures, and hypothetical real-life characters. The profiles of the first two

---

*Corresponding authors: Xu Chen (xu.chen@ruc.edu.cn) and Zhiwu Lu (luzhiwu@ruc.edu.cn).
[1]https://github.com/YanqiDai/MMRole

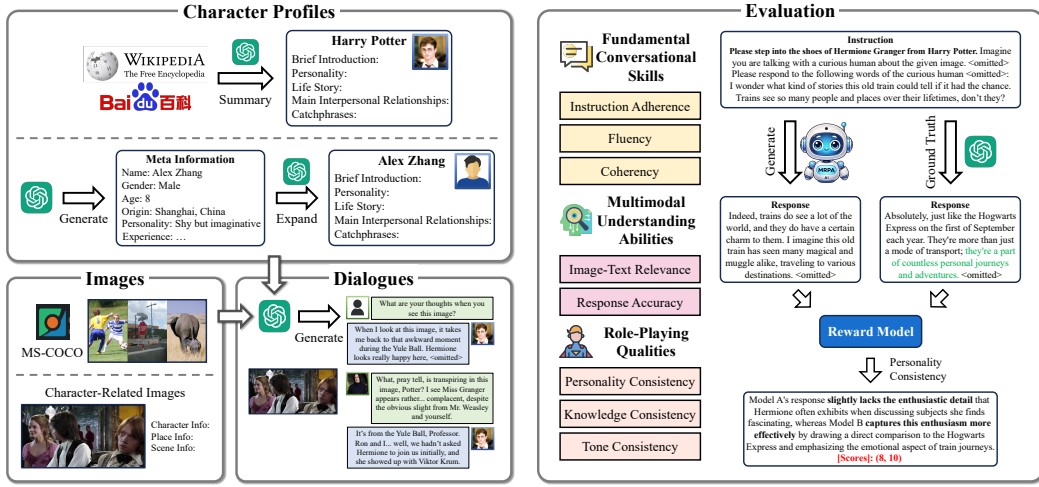

Figure 1: An overview of the *MMRole* framework. (a) *MMRole-Data* includes character profiles, images, and dialogues centered around images. (b) *MMRole-Eval* comprises eight evaluation metrics across three dimensions. For each metric, the reward model scores MRPAs with the constructed ground-truth data for comparison.

categories are summarized by GPT-4 (Achiam et al., 2023) based on information from Wikipedia or Baidu Baike, while those of the last category are randomly generated by GPT-4. For each character, we utilize distinct generic images from MS-COCO (Lin et al., 2014), and manually collect and annotate various character-related images. Finally, the dialogues are generated by GPT-4 based on the character profiles and images, occurring either between a character and a human user or between two characters. Both the character profiles and the dialogues are subjected to rigorous manual quality control to ensure accuracy and reliability. Statistically, the *MMRole-Data* dataset comprises 85 characters, 11K images, and 14K dialogues, yielding 85K training samples and 294 test samples.

**Performance Evaluation:** On one hand, MRPAs must not only accurately emulate specific characters but also deeply comprehend both visual and textual information. On the other hand, existing methods for evaluating RPAs directly score outputs without a ground truth (Zhou et al., 2023a; Tu et al., 2024), which may lead to unstable scoring criteria without a baseline for comparison. Therefore, we propose *MMRole-Eval*, a robust evaluation approach to stably and comprehensively assess MRPAs, which comprises eight metrics across three dimensions: fundamental conversational skills, multimodal understanding abilities, and role-playing qualities. For each metric, our specialized reward model initially conducts a brief qualitative assessment of the relative performance between the evaluated MRPA and the constructed ground-truth data, followed by assigning a quantitative score pair. The final score of the MRPA is the ratio of the two scores within the score pair. To develop the reward model, we employ GPT-4 to assess various MRPAs and leverage the evaluation trajectories to train our reward model, which renders *MMRole-Eval* both open-source and cost-effective.

Briefly, our main contributions are three-fold:

1. We propose the concept of Multimodal Role-Playing Agents (MRPAs) for the first time, and construct *MMRole-Data*, a large-scale, high-quality dataset for developing and evaluating MRPAs.

2. We introduce *MMRole-Eval*, a robust evaluation approach to stably and comprehensively assess MRPAs, comprising eight metrics across three dimensions. A specialized reward model is trained to score MRPAs with the constructed ground-truth data for comparison.

3. We develop the first specialized MRPA, *MMRole-Agent*, and conduct comprehensive evaluations and analyses of *MMRole-Agent* alongside various general-dialogue large multimodal models.

## 2 RELATED WORK

**Role-Playing Agents.** Recent advancements in large language models (LLMs) (Zhao et al., 2023), such as supervised fine-tuning (Wei et al., 2021) and in-context learning (Brown et al., 2020), have significantly catalyzed the rise of Role-Playing Agents (RPAs) (Chen et al., 2024b), which are in-

teractive AI systems that can emulate designated personas. Specifically, the personas can be categorized into individual characters (Wang et al., 2023; Shao et al., 2023; Wang et al., 2024c; Gu et al., 2024) and groups of people with particular attributes (Li et al., 2023b; Hong et al., 2023; Xu et al., 2023; Zhang et al., 2024). In this study, we primarily focus on the former.

Existing RPAs that emulate individual characters are developed through either training or prompting LLMs with high-quality character-specific dialogues. In a pioneering study, Chen et al. (2023) extracted all dialogue sessions from original scripts to develop a Harry Potter-specific RPA. Furthermore, Wang et al. (2023), Zhou et al. (2023a), Shao et al. (2023) and Li et al. (2023a) constructed hundreds of characters and more comprehensive datasets of character dialogues. These efforts aimed to develop RPAs for delivering emotional value to humans. In contrast, Gu et al. (2024) focused on facilitating sociological research. However, these studies are primarily confined to the textual modality. Conversely, our *MMRole* framework is the first to enhance RPAs with multimodal capabilities.

The evaluation of RPAs is also a crucial and challenging research direction. Diverse methods have been proposed. Specifically, Shen et al. (2023) and Chen et al. (2024a) assessed RPAs with multiple-choice questions. Tu et al. (2024) trained a reward model for scoring without a ground truth. Wang et al. (2024d) evaluated the personality fidelity of RPAs through interviews, scoring without a ground truth by GPT-4. Ng et al. (2024) engaged the acquaintances of the target individuals to distinguish between humans and RPAs. Wang et al. (2024a) further evaluated RPAs in text-based virtual worlds. However, the high expense of human annotation and the potential instability of scoring without a ground truth pose significant challenges. To address this, we develop a reward model to score RPAs with a ground-truth baseline for comparison.

**Large Multimodal Models.** Large Multimodal Models (LMMs) are advanced AI systems typically built upon LLMs, designed to integrate and comprehend multiple data modalities, particularly text and images (Yin et al., 2023). A variety of impressive LMMs have been released, including closed-source models with hundreds of billions of parameters like GPT-4V (Achiam et al., 2023), Gemini (Team et al., 2023), and Claude 3 (Anthropic, 2024), and open-source models with tens of billions or billions of parameters like MiniGPT-4 (Zhu et al., 2023), InstructBLIP (Dai et al., 2023), LLaVA (Liu et al., 2024c;a;b; Li et al., 2024a), QWen-VL (Bai et al., 2023; Wang et al., 2024b), InternVL (Chen et al., 2024c), and Yi-VL (Young et al., 2024). Additionally, various techniques have been explored to enhance the performance of LMMs, such as visual instruction tuning (Liu et al., 2024c), mixture of experts (Lin et al., 2024), and multi-task balancing (Dai et al., 2024). LMMs are widely applied in vertical fields, including healthcare (Li et al., 2024b), document understanding (Ye et al., 2023), and GUI navigation (Hong et al., 2024). To further explore their potential, we apply LMMs to role-playing for the first time.

## 3 MULTIMODAL ROLE-PLAYING AGENTS

Role-Playing Agents (RPAs) are engineered to emulate specific characters and engage in dialogues with either human users or other characters. Expanding on this concept, Multimodal Role-Playing Agents (MRPAs) incorporate the capacity to comprehend vision-language multimodal information. This capacity enables dialogues that are centered around and informed by images. From another perspective, compared to traditional multimodal question answering, multimodal role-playing includes character profile input, adding greater complexity and depth to the interaction.

In scenarios where the dialogue partner is a human user without a specific identity, given an image $I$, the profile $P$ of the designated character $C$, and the dialogue context $D$, the MRPA steps into the shoes of the character $C$, responding to the human user about the image $I$:

$$D = [h_1, m_1, h_2, m_2, \ldots, h_n], \tag{1}$$

$$m_n = \text{MRPA}(I, P, D), \tag{2}$$

where $D$ is a sequence of conversation pairs, with $h_i$ and $m_i$ representing the $i$-th utterances from the human user and the MRPA, respectively.

Conversely, in scenarios where the dialogue partner is another character $C_{\text{other}}$, given an image $I$, the profile $P$ of the designated character $C$, the profile $P_{\text{other}}$ of the character $C_{\text{other}}$, and the dialogue context $D$, the MRPA steps into the shoes of the character $C$ and interacts with the character $C_{\text{other}}$, either initiating or responding within the dialogue centered around the image $I$:

$$D = [o_1, m_1, o_2, m_2, \ldots, o_n] \text{ or } [m_1, o_1, m_2, o_2, \ldots, m_{n-1}, o_{n-1}], \tag{3}$$

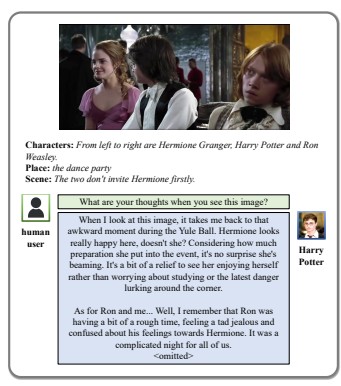
(a) Commentary Interaction

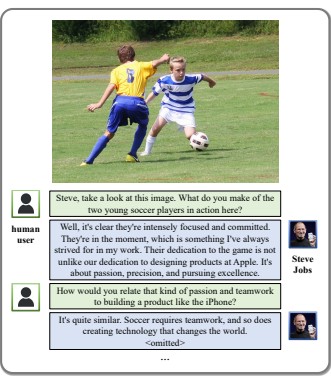
(b) Human-Role Dialogue

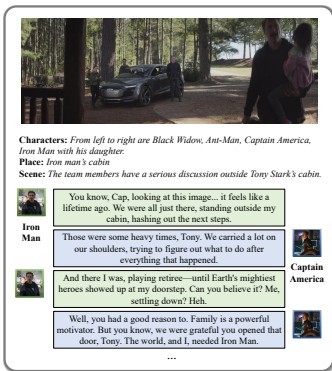
(c) Inter-Role Dialogue

Figure 2: Examples of the three types of dialogue scenarios in *MMRole-Data*.

$$m_n = \text{MRPA}(I, P, P_{\text{other}}, D), \qquad (4)$$

where $D$ is a sequence of conversation pairs, with $o_i$ and $m_i$ signifying the $i$-th utterances from the character $C_{\text{other}}$ and the MRPA, respectively. Notably, both the character $C_{\text{other}}$ and the MRPA can potentially initiate the dialogue.

## 4 *MMRole-Data*: DATASET CONSTRUCTION

As shown in 1(a), we construct *MMRole-Data*, a large-scale, high-quality multimodal role-playing dataset. In this section, we first provide a detailed classification of characters and dialogue scenarios considered in *MMRole-Data*, then describe the pipelines for character profile generation and image collection and annotation, as well as the methodology for dialogue generation and filtering.

### 4.1 CHARACTERS AND DIALOGUE SCENARIOS

We consider three categories of characters: **(1) Fictional Characters**, characters created in fictional media such as literature, films, and games; **(2) Historical and Public Figures**, individuals who are specifically documented in historical records or well-known in real life; **(3) Hypothetical Real-Life Characters**, hypothetical individuals who are not explicitly known but could exist in real life.

The first two categories have been explored in previous role-playing research. Moreover, we propose the third category to enhance and evaluate MRPAs in characters that are not widely recognized. To effectively emulate hypothetical real-life characters, MRPAs must deeply understand and align with the provided character profiles, rather than relying on their inherent world knowledge.

As depicted in Figure 2, we introduce three types of dialogue scenarios consistently centered around images: **(1) Commentary Interactions**, single-turn dialogues where a character offers comments or reflections centered around an image, without any further interaction; **(2) Human-Role Dialogues**, multi-turn dialogues centered around an image between a human user without a specific identity and a character; **(3) Inter-Role Dialogues**, multi-turn dialogues centered around an image between two characters from the same series.

### 4.2 CHARACTER PROFILE GENERATION

Character profiles are crucial for the role-playing effectiveness of MRPAs, especially for those characters with which MRPAs are not familiar. To facilitate a thorough understanding of the designated characters, our character profiles encompass five core parts: brief introduction, personality, life story, main interpersonal relationships, and catchphrases, which are exampled in Appendix D.

As discussed in Section 4.1, three categories of characters are considered in *MMRole*. The majority of these characters are English, with a smaller proportion being Chinese. For fictional characters, as well as historical and public figures, the profiles are summarized by GPT-4 where the information is sourced from Wikipedia for English characters and Baidu Baike for Chinese characters. Furthermore, for hypothetical real-life characters, the profiles are generated through a two-stage process by GPT-4 to ensure both universality and diversity. Firstly, GPT-4 generates the meta information for all characters in a single API call, including basic details such as names and genders, and brief

descriptions of personalities and backgrounds. We instruct GPT-4 that "*The character information should cover as many different situations as possible to reflect the diversity and complexity of human society*". Secondly, GPT-4 expands the meta information of each character to derive the profile. The two-stage generation process is exemplified in Appendix E. Additionally, all character profiles undergo rigorous manual quality control to ensure accuracy and reliability, detailed in Appendix F, and are simplified by GPT-4 to adhere to the context length limits of most LMMs.

### 4.3 IMAGE COLLECTION AND ANNOTATION

For each character, we utilize distinct generic images from MS-COCO (Lin et al., 2014) to ensure comprehensive coverage of a wide range of visual concepts. Additionally, we manually collect and annotate various character-related images, which can evoke the personal experiences and emotions of the characters more effectively. Specifically, we collect production stills for fictional characters, web illustrations for historical and public figures, and news photos for hypothetical real-life characters. Moreover, as presented in Figure 2, the information of characters, place, and scene is manually annotated for each character-related image.

### 4.4 DIALOGUE GENERATION AND FILTERING

As discussed in Section 4.1, three types of dialogue scenarios are introduced in *MMRole*. Based on the character profiles and images, GPT-4 generates dialogues corresponding to each scenario type. Interestingly, we observe that using the prompt, "*You are a dedicated role-playing assistant...Please step into the shoes of {character} from {series}*" yields better results than the simpler prompt, "*You are {character} from {series}*". We suggest that the training data supplied by OpenAI optimizes GPT-4 to function more effectively as a helpful assistant, rather than as an immersive, human-like character. The prompts for dialogue generation are detailed in Appendix I. To ensure accuracy and reliability, we manually filter all dialogues using several strategies, detailed in Appendix F.

## 5 *MMRole-Eval*: PERFORMANCE EVALUATION

As illustrated in Figure 1(b), we propose *MMRole-Eval*, a robust evaluation approach to stably and comprehensively assess MRPAs. In this section, we introduce eight evaluation metrics across three dimensions and the approach for score quantification.

### 5.1 EVALUATION METRICS

In contrast to textual RPAs, MRPAs must not only accurately emulate specific characters but also deeply comprehend both visual and textual information. Therefore, we propose a three-dimensional evaluation system, encompassing fundamental conversational skills, multimodal understanding abilities, and role-playing qualities.

The fundamental conversational skills of MRPAs present their capacity to sustain fluent and coherent interactions within role-playing scenarios, which are assessed by three metrics:

- **Instruction Adherence (IA)**: Do the responses accurately adhere to the task instruction, directly role-playing as the character and including only words that the character would say, without any unnecessary explanatory prefixes or suffixes?
- **Fluency (Flu)**: Are the responses grammatically correct and articulated smoothly?
- **Coherency (Coh)**: Do the responses maintain a coherent thread of dialogue without contradicting previous turns or containing internal inconsistencies within the current responses?

The multimodal understanding abilities of MRPAs indicate their capacity to effectively integrate and interpret both visual and textual information, which are assessed by two metrics:

- **Image-Text Relevance (ITR)**: Do the responses exhibit a close correlation with the visual content depicted in the image?
- **Response Accuracy (RA)**: Do the responses accurately answer the words of the human user or the other character, or appropriately initiate a conversation based on the image?

The role-playing qualities of MRPAs denote their capacity to convincingly emulate characters, maintaining consistency in personality, knowledge, and tone, which are assessed by three metrics:

- **Personality Consistency (PC)**: Do the responses accurately and deeply reflect the personality of the character?

Table 1: The statistics of *MMRole-Data*. 'CR Images' represents character-related images. 'In-Test' denotes the in-distribution test set, while 'Out-Test' signifies the out-of-distribution test set.

|  | Train | In-Test | Out-Test | Overall |
|---|---|---|---|---|
| Characters | 72 | | 13 | 85 |
| Generic Images | 10,800 | | 39 | 10,839 |
| CR Images | 175 | | 18 | 193 |
| Dialogues | 14,052 | 216 | 78 | 14,346 |
| Samples | 85,456 | 216 | 78 | 85,750 |

Table 2: The statistics for the three types of dialogue scenarios in *MMRole-Data*.

|  | Comment. | Human-Role. | Inter-Role. | Overall |
|---|---|---|---|---|
| Dialogues | 4893 | 4617 | 4836 | 14346 |
| Turns / Dlg. | 1.00 | 5.80 | 5.75 | 4.15 |
| Tokens / Dlg. | 236.00 | 446.91 | 429.54 | 369.12 |

- **Knowledge Consistency (KC)**: Do the responses accurately reflect the knowledge of the character, encompassing their experiences, abilities, and relationships?
- **Tone Consistency (TC)**: Do the responses align with the typical speech patterns and catchphrases of the character, rather than resembling the style of AI assistants?

## 5.2 SCORE QUANTIFICATION

To quantitatively evaluate the performance of RPAs across various metrics, existing methods utilize reward models or human annotators to directly score outputs without a ground truth (Zhou et al., 2023a; Tu et al., 2024). However, it may be unstable due to the variability of scoring criteria without a baseline for comparison. Therefore, we propose to develop a more stable reward model. Inspired by the evaluation methods of Vicuna (Chiang et al., 2023) and LLaVA (Liu et al., 2024c), our reward model first conducts a brief qualitative assessment of the relative performance between the evaluated MRPA and the constructed ground-truth data for each metric, followed by assigning a quantitative score pair. The final score of the MRPA is the ratio of the two scores within the score pair.

To develop the reward model, we initially employ GPT-4 to assess various MRPAs across all test samples. For each evaluated MRPA and corresponding test sample, GPT-4 outputs brief assessments and score pairs for all metrics through a single API call. Subsequently, these evaluation trajectories are converted into training and validation data for our reward model. Each evaluation trajectory is segmented into eight samples, with each sample evaluating a distinct metric. The prompts for GPT-4 scoring and reward model scoring are provided in Appendix H. Compared to directly applying GPT-4 as the reward model, this approach renders *MMRole-Eval* both open-source and cost-effective.

## 6 EXPERIMENTS

### 6.1 STATISTICS OF *MMRole-Data*

Table 1 presents the statistics of the *MMRole-Data* dataset. Totally, the dataset comprises 85 characters, 11,032 images, and 14,346 dialogues, yielding 85,456 training samples and 294 test samples. Specifically, we construct 72 characters and collect 10,975 images for training and in-distribution testing. For out-of-distribution testing, we additionally construct 13 characters and collect 57 images that differ from those used in the former set. Dividing the data in this manner is significant for evaluating the performance of MRPAs on previously unseen characters and images, thereby assessing its generalization capabilities. All constructed characters are listed in Appendix C.

From another perspective, Table 2 illustrates the statistics for the three types of dialogue scenarios in the *MMRole-Data* dataset. The commentary interactions are single-turn, whereas both the human-role dialogues and the inter-role dialogues involve multiple turns. When converting dialogues into training and test samples, a single multi-turn dialogue entry can generate multiple training samples or a single test sample randomly selected from a specific turn.

Table 3: The evaluated MRPAs in our experiments, which are grouped by parameter scale.

| MRPAs | Version | Params | Open-Source | Specialized |
|---|---|---|---|---|
| GPT-4 Turbo (Achiam et al., 2023) | 2024-04-09 | > 100B | ✗ | ✗ |
| Gemini Pro Vision (Team et al., 2023) | 2023-12-13 | > 100B | ✗ | ✗ |
| Claude 3 Opus (Anthropic, 2024) | 2024-02-29 | > 100B | ✗ | ✗ |
| QWen-VL-Max (Bai et al., 2023) | 2023-12-01 | > 100B | ✗ | ✗ |
| LLaVA-NeXT-34B (Liu et al., 2024b) | 2024-01-30 | 34B | ✓ | ✗ |
| Yi-VL-34B (Young et al., 2024) | 2024-01-23 | 34B | ✓ | ✗ |
| InternVL-Chat-V1.5 (Chen et al., 2024c) | 2024-04-18 | 26B | ✓ | ✗ |
| QWen-VL-Chat (Bai et al., 2023) | 2023-08-22 | 9B | ✓ | ✗ |
| LLaVA-NeXT-Mistral-7B (Liu et al., 2024b) | 2024-01-30 | 7B | ✓ | ✗ |
| Yi-VL-6B (Young et al., 2024) | 2024-01-23 | 6B | ✓ | ✗ |
| *MMRole-Agent* (ours) | | 9B | ✓ | ✓ |

Table 4: The validation mean absolute error (MAE) results for the effectiveness of the reward model. 'QWen-VL-Chat (GPT-4)' and 'Reward Model (GPT-4)' denote the scores evaluated by QWen-VL-Chat and the reward model compared to those evaluated by GPT-4. 'QWen-VL-Chat (humans)', 'GPT-4 (humans)', and 'Reward Model (humans)' signify the score gaps provided by QWen-VL-Chat, GPT-4, and the reward model compared to the ground-truth score gaps provided by humans.

| Evaluators (Ground Truth) | IA | Flu | Coh | ITR | RA | PC | KC | TC | Overall |
|---|---|---|---|---|---|---|---|---|---|
| QWen-VL-Chat (GPT-4) | 0.3776 | 0.3718 | 0.3218 | 0.3561 | 0.3528 | 0.4091 | 0.3794 | 0.4558 | 0.3780 |
| Reward Model (GPT-4) | 0.0708 | 0.0387 | 0.0526 | 0.0568 | 0.0584 | 0.1165 | 0.0815 | 0.1154 | 0.0738 |
| QWen-VL-Chat (humans) | 0.2469 | 0.1870 | 0.2720 | 0.2574 | 0.2608 | 0.2368 | 0.2243 | 0.2658 | 0.2439 |
| GPT-4 (humans) | 0.1526 | 0.1150 | 0.0772 | 0.0922 | 0.1463 | 0.1475 | 0.1279 | 0.1442 | 0.1254 |
| Reward Model (humans) | 0.0993 | 0.0815 | 0.1006 | 0.1225 | 0.1412 | 0.1669 | 0.1438 | 0.1507 | 0.1258 |

## 6.2 DEVELOPMENT OF *MMRole-Agent*

We fine-tune the QWen-VL-Chat model (Bai et al., 2023) using $8\times$A100 GPUs on the training set of *MMRole-Data* to develop our specialized MRPA, *MMRole-Agent*. Integrating data from different characters and dialogue scenarios for multi-task training can improve the generalization capabilities of *MMRole-Agent*. The learning rate is set to $1e-5$, and the training is conducted over 3 epochs. To accommodate detailed character profiles and dialogue history, the model maximum length is set to 3072. Other experimental setups and codes remain the same as Bai et al. (2023)'s defaults.

## 6.3 EVALUATED MRPAS

To the best of our knowledge, no specialized MRPA has been developed prior to this work. Therefore, our experiments evaluate *MMRole-Agent* and various existing general-dialogue LMMs across different parameter scales. As presented in Table 3, we select four well-known closed-source LMMs with over 100 billion parameters (Achiam et al., 2023; Team et al., 2023; Anthropic, 2024; Bai et al., 2023), and six widely-used open-source LMMs with tens of billions or billions of parameters (Liu et al., 2024b; Bai et al., 2023; Young et al., 2024; Chen et al., 2024c). For the closed-source models, we utilize their official APIs to conduct performance evaluations. To ensure fairness, each MRPA is queried with the same prompt, as detailed in Appendix G.

## 6.4 DEVELOPMENT AND VALIDATION OF THE REWARD MODEL

To develop the reward model, we initially utilize GPT-4 to evaluate various general-dialogue LMMs discussed in Section 6.3 across 294 test samples. Statistically, these evaluation trajectories are converted into a total of 23,520 samples, where 320 samples are reserved for validation, with the rest utilized for training. The validation set includes 20 questions, where the responses of two models are randomly selected for each question, and each response is evaluated on all 8 metrics. Subsequently, another QWen-VL-Chat model (Bai et al., 2023) is trained to develop the specialized reward model.

The experimental setup and code are the same as those used for developing *MMRole-Agent*, except that the model maximum length is set to 4096, and the training is conducted over 10 epochs.

The scoring success rate of the base model QWen-VL-Chat is only 33.13%, whereas those of the reward model and GPT-4 are both 100%. To further validate the effectiveness of the reward model, we initially calculate the mean absolute errors (MAEs) of the scores evaluated by QWen-VL-Chat and the reward model compared to those evaluated by GPT-4. The score is random if the model fails to score. As illustrated in Table 4, the overall MAE of QWen-VL-Chat (GPT-4) is remarkably high, whereas that for Reward Model (GPT-4) is less than 0.1. Furthermore, we engage four human evaluators to compare responses from two MRPAs on each metric for every question in the validation set. Their choices among 'better', 'equal', and 'worse' correspond to the score gaps of 0.4, 0, and −0.4 between the two responses, respectively. The results from all evaluators are averaged to obtain the ground-truth score gaps. In this manner, it is easier for human evaluators to yield more consistent results among individuals than directly scoring the MRPA's responses. Subsequently, we calculate the

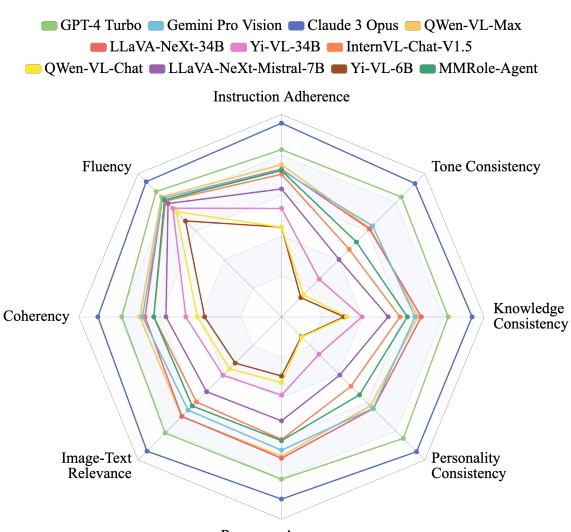

Figure 3: The visualization of the evaluation results for all MRPAs. Each indicator displays an interval length of 0.55, and the maximum value of the interval for different indicators is adjusted from 1.10 to 1.25.

MAEs between the score gaps provided by QWen-VL-Chat, GPT-4, and the reward model compared to those provided by human evaluators. In this context, score gaps are capped at 0.4 and -0.4. As shown in Table 4, the overall MAE for QWen-VL-Chat (humans) is significantly high, whereas those for GPT-4 (humans) and Reward Model (humans) are comparable and considerably low. Moreover, we report the root mean squared error (RMSE) and Pearson correlation coefficient (Pearson) results in Appendix J. These results indicate that our specialized reward model effectively learns the evaluation abilities of GPT-4 and aligns closely with human evaluators, significantly superior to the non-specialized QWen-VL-Chat model.

Besides, we assess the internal consistency of *MMRole-Eval* by Cronbach's alpha coefficient (Cronbach, 1951), where the score of an evaluated MRPA for a given test query is computed as the average across all metrics. The resulting value of 0.70 indicates that *MMRole-Eval* exhibits a moderate level of internal consistency. The relatively lower value can be attributed to the diversity of characters and dialogue scenarios, as individual queries inherently assess distinct aspects.

## 6.5 EVALUATION RESULTS AND ANALYSES

As shown in Table 5, we report the average results across all test samples for each evaluated MRPA, along with the detailed results on both the in-distribution test set (In-Test) and the out-of-distribution test set (Out-Test) for our *MMRole-Agent*. Notably, although some of the scores of MRPAs exceed 1, this does not necessarily mean that their performance is superior to that of GPT-4, which we use to construct *MMRole-Data*. To generate multi-turn dialogue data, we use a single GPT-4 API call to produce the entire dialogue directly. In contrast, when testing MRPAs, we supply dialogue histories and require MRPAs to generate responses. This approach is relatively easier, but it is challenging to ensure the consistency of multi-turn dialogues if used for data construction.

In the MRPA group with over 100 billion parameters, Claude 3 Opus exhibits superior performance. Meanwhile, in the MRPA group with tens of billions of parameters, LLaVA-NeXT-34B achieves the highest performance. Finally, in the MRPA group with billions of parameters, *MMRole-Agent* is the best. Notably, LLaVA-NeXT-34B outperforms Gemini Pro Vision, while LLaVA-NeXT-Mistral-7B and *MMRole-Agent* surpass Yi-VL-34B. This suggests that both the training methods and training data are important for enhancing LMMs, rather than merely expanding the model size.

Table 5: The average results across all test samples for each evaluated MRPA, along with the detailed results for our *MMRole-Agent* on both the in-distribution test set (In-Test) and the out-of-distribution test set (Out-Test). In each group categorized by parameter scale, the best overall result is **bolded**, while the second-best one is underlined.

| MRPAs | IA | Flu | Coh | ITR | RA | PC | KC | TC | Overall |
|---|---|---|---|---|---|---|---|---|---|
| GPT-4 Turbo | 1.055 | 1.032 | 1.084 | 1.097 | 1.092 | 1.168 | 1.103 | 1.161 | 1.099 |
| Gemini Pro Vision | 0.999 | 1.007 | 1.028 | 1.009 | 1.013 | 1.052 | 1.013 | 1.050 | 1.021 |
| Claude 3 Opus | 1.127 | 1.070 | 1.149 | 1.167 | 1.146 | 1.219 | 1.168 | 1.213 | **1.157** |
| QWen-VL-Max | 1.014 | 1.012 | 1.035 | 1.034 | 1.029 | 1.042 | 1.021 | 1.041 | 1.028 |
| LLaVA-NeXT-34B | 1.002 | 1.007 | 1.021 | 1.033 | 1.035 | 1.053 | 1.030 | 1.038 | **1.027** |
| Yi-VL-34B | 0.895 | 0.968 | 0.910 | 0.875 | 0.863 | 0.844 | 0.869 | 0.845 | 0.884 |
| InternVL-Chat-V1.5 | 0.988 | 0.996 | 0.997 | 0.977 | 0.984 | 0.967 | 0.972 | 0.960 | 0.980 |
| QWen-VL-Chat | 0.844 | 0.954 | 0.879 | 0.850 | 0.829 | 0.778 | 0.827 | 0.785 | 0.843 |
| LLaVA-NeXT-Mistral-7B | 0.948 | 0.986 | 0.964 | 0.938 | 0.933 | 0.924 | 0.940 | 0.921 | 0.944 |
| Yi-VL-6B | 0.844 | 0.919 | 0.859 | 0.828 | 0.811 | 0.776 | 0.820 | 0.774 | 0.829 |
| *MMRole-Agent* | 0.998 | 1.000 | 0.997 | 0.993 | 0.987 | 1.000 | 0.992 | 0.988 | **0.994** |
| *MMRole-Agent* (In-Test) | 1.000 | 1.000 | 0.999 | 0.997 | 0.989 | 1.012 | 0.997 | 0.997 | 0.999 |
| *MMRole-Agent* (Out-Test) | 0.992 | 0.999 | 0.993 | 0.979 | 0.981 | 0.963 | 0.977 | 0.962 | 0.981 |

Table 6: The average results across all test samples for each evaluated RPAs. 'w/o vision' signifies that image information is excluded from the input prompt of RPAs.

| RPAs | Comment. | Human-Role. | Inter-Role. | Overall |
|---|---|---|---|---|
| GPT-4 Turbo w/o vision | 0.5746 | 1.1330 | 1.0843 | 0.9306 |
| GPT-4 Turbo | 1.0261 | 1.2275 | 1.3450 | 1.1995 |
| Claude 3 Opus w/o vision | 0.3290 | 1.1803 | 1.1420 | 0.8838 |
| Claude 3 Opus | 1.0088 | 1.2889 | 1.3916 | 1.2298 |
| *MMRole-Agent* w/o vision | 0.4192 | 0.8909 | 0.7907 | 0.7003 |
| *MMRole-Agent* | 1.0450 | 0.9556 | 0.9619 | 0.9875 |

Moreover, the overall score of *MMRole-Agent* reaches $0.994$, marking a significant improvement of $0.151$ compared to its base model, QWen-VL-Chat. *MMRole-Agent* successfully acquires various capabilities required for the MRPA from *MMRole-Data*, outperforming all evaluated open-source LMMs, except for LLaVA-NeXT-34B. Besides, *MMRole-Agent* achieves similar overall scores on both the in-distribution test set and the out-of-distribution test set, with the latter being slightly lower by $0.018$. This indicates that *MMRole-Agent* has strong generalization capabilities for characters and images that are not seen in the training set.

As shown in Figure 3, we provide a clear visual representation of the evaluation results. The overall performance rankings of MRPAs closely align with their specific rankings on each metric. However, significant differences exist in the score variations across various metrics. Specifically, all MRPAs achieve high scores on the Fluency metric with minimal variations, suggesting that producing fluent content is not a major challenge for current LMMs. Conversely, there are notable differences among MRPAs on other metrics, particularly on Personality Consistency and Tone Consistency. It reveals that multimodal understanding abilities and role-playing qualities are more challenging aspects that require attention in the development of MRPAs.

Additionally, to highlight the inherent advantages of MRPAs over single-modal RPAs, we conduct comparative experiments on two SOTA general-purpose LMMs and our *MMRole-Agent*. As presented in Table 6, we report the Image-Text Relevance scores on the Out-Test set evaluated by GPT-4, where 'w/o vision' signifies that image information is excluded from the input prompt of RPAs. The results clearly demonstrate that excluding image information significantly reduces the Image-Text Relevance of all RPAs' responses, particularly in commentary interaction scenarios. In multi-turn human-role and inter-role dialogue scenarios, textual dialogue history could provide in-

Table 7: The results on the Out-Test set for *MMRole-Agent* with different numbers of characters.

| Characters | Number of Characters | Overall |
|---|---|---|
| The Avengers | 16 | 0.965 |
| The Avengers + Other English Fictional Characters | 34 | 0.968 |
| The Avengers + Hypothetical Real-Life Characters | 36 | 0.970 |
| ALL (ours) | 72 | **0.983** |

direct clues about images, leading to relatively smaller declines in the Image-Text Relevance scores compared to those in commentary interactions. Nevertheless, the absence of visual inputs still results in a marked drop in performance across all scenarios.

### 6.6 DETAILED ANALYSES OF *MMRole-Agent*

To demonstrate the superiority of our *MMRole-Agent*, use cases of *MMRole-Agent*, GPT-4, and QWen-VL-Chat are presented and analyzed in Appendix K. Moreover, sensitivity test results detailed in Appendix L indicate that *MMRole-Agent* is compatible with different prompts and does not exhibit signs of overfitting. The strong performance and generalization abilities of our *MMRole-Agent* can be primarily ascribed to the following two factors:

Table 8: The average results across all test samples for *MMRole-Agent* with different numbers of training samples.

| Training Data | Number of Samples | Overall |
|---|---|---|
| SAMPLE | 8.5K | 0.967 |
| ALL (ours) | 85K | **0.989** |

1. **Training with Large-Scale, High-Quality Data:** The training set of *MMRole-Data* comprises 72 characters, 11K images, and over 85K samples. Furthermore, as depicted in Figure 1(a) and Figure2, due to the well-designed data construction pipeline, meticulous manual annotation and quality control, and the use of GPT-4, the data is of high quality. This large-scale, high-quality dataset enables *MMRole-Agent* to comprehensively learn the instruction demands, knowledge, and abilities in multimodal role-playing. To verify this point, we compare the performance differences between a model trained on the full dataset (ALL) and a model trained on a randomly sampled subset consisting of one-tenth of the data (SAMPLE), both evaluated after one epoch of training. As shown in Table 8, the performance of ALL is superior to that of SAMPLE.

2. **Joint Training with Diverse Multi-Character Data:** We incorporate data from 72 diverse characters to jointly train a unified *MMRole-Agent*. This approach, akin to the principles of multi-task learning, enables the model to acquire generalizable multimodal role-playing capabilities, rather than being confined to specific characters. To verify this point, we first train a model using data of characters from The Avengers, then gradually add additional characters to the training set for subsequent models. As presented in Table 7, we evaluate the performance of each model on the Out-Test set. The models' zero-shot performance steadily improves as more characters are incorporated. Notably, with comparable numbers of characters, introducing hypothetical real-life characters (with significant differences from The Avengers) yields greater gains than adding other English fictional characters, indicating the significance of training with diverse data.

## 7 CONCLUSION

In this paper, we propose the concept of Multimodal Role-Playing Agents (MRPAs) for the first time by extending RPAs with multimodal understanding abilities. Moreover, we construct *MMRole-Data*, a large-scale, high-quality dataset for developing and evaluating MRPAs. To stably and comprehensively assess MRPAs, we introduce *MMRole-Eval*, a robust evaluation approach that comprises eight metrics across three dimensions, scoring MRPAs with the ground truth for comparison by a specialized reward model. Evaluation results reveal that our *MMRole-Agent*, the first specialized MRPA, exhibits improved performance and strong generalization capabilities. Additionally, multimodal understanding abilities and role-playing qualities are more challenging aspects that require attention in the development of MRPAs. However, there exists a limitation that the training data for *MMRole-Agent* is primarily synthesized by GPT-4, which constrains its performance from surpassing GPT-4 itself. In future work, we will address this limitation by leveraging multiple SOTA LMMs respectively as responders, reviewers, and summarizers, striving to push the boundaries of its capabilities.

ACKNOWLEDGMENTS

This work was supported in part by National Natural Science Foundation of China (62437002, 62376274) and Beijing Natural Science Foundation (L233008).

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

## A   Ethics Statement

This work adheres to the ICLR Code of Ethics, ensuring ethical compliance throughout all stages of the research. The *MMRole-Data* dataset was constructed using publicly available data, with rigorous quality control to prevent privacy risks. We acknowledge the potential biases present in the data and have taken proactive measures to ensure diversity and reduce these biases. Moreover, we recognize that MRPAs may generate dialogues that could misleadingly appear as actual statements made by real individuals. To prevent misunderstanding, it is essential to explicitly indicate that such content is simulated and does not represent genuine speech.

## B   Reproducibility Statement

For research reproducibility, the data, code, and models are all available at this GitHub repository. Additionally, the training settings of *MMRole-Agent* and the reward model in *MMRole-Eval* are presented in Section 6.2 and Section 6.4, respectively, while the training and inference code, along with the detailed assessment results of all evaluated MRPAs are submitted as supplementary materials. Moreover, the detailed prompts for dataset construction and performance evaluation are provided in Appendix E, Appendix I, Appendix G, and Appendix H, while the generated data is exemplified in Appendix C, Appendix D, and Figure 2.

## C   List of Characters

---

**Fictional Characters**

*The Avengers*: Iron Man, Captain America, Thor, Hulk, Spider-Man, Black Widow, Hawkeye, Loki, Doctor Strange, Vision, Black Panther, Ant-Man, Scarlet Witch, Star-Lord, Nick Fury, Thanos
*X-Men*: Wolverine, Professor X, Magneto, Phoenix
*Harry Potter*: Harry Potter, Hermione Granger, Ron Weasley, Albus Dumbledore, Severus Snape, Lord Voldemort
*Toy Story*: Woody, Buzz Lightyear
Friends: Rachel Green, Monica Geller, Phoebe Buffay, Joey Tribbiani, Chandler Bing, Ross Geller
*西游记*: 孙悟空, 唐僧, 猪八戒, 沙悟净
*成龙历险记*: 成龙, 小玉, 老爹
*The Big Bang Theory*: Howard Wolowitz, Leonard Hofstadter, Penny, Raj Koothrappali, Sheldon Cooper

**Historical and Public Figures**

*the Renaissance period*: Michelangelo, Leonardo da Vinci, William Shakespeare, Nicolaus Copernicus
*the modern times*: Stephen Hawking, Steve Jobs, Kobe Bryant, Diego Maradona, Michael Jackson
*中国唐代*: 杜甫, 李白
*the Ancient Greece period*: Socrates, Plato, Aristotle

**Hypothetical Real-Life Characters**

*Hypothetical Characters*: Maya Patel, Liam Johnson, Sofia Rodriguez, Takumi Nakamura, Chloe Dubois, Henry O'Malley, Aisha Al-Farsi, Carlos Rivera, Nia Williams, Alex Zhang, Elizabeth "Lizzy" Thompson, Dimitri Petrov, Jasmine Lee, Michael O'Reilly, Sunita Krishnan, Luca Bianchi, Fatima Zahra, Ethan Wright, Priya Singh, Omar Abdullah
*Hypothetical Characters*: Javier Martinez, Ayesha Khan, Timothy Clark, Elena Petrova, Charles "Charlie" Wembly

---

Figure 4: All character constructed in *MMRole-Data*, with the series to which the characters in the out-of-distribution test set belong being underlined.

Figure 4 lists all characters constructed in *MMRole-Data*, with the series to which the characters in the out-of-distribution test set belong being underlined.

## D   Examples of Character Profiles

Figure 5 presents the profile of Iron Man from *The Avengers*, whereas Figure 6 illustrates the profile of Li Bai from *Tang Dynasty of China*. The character profiles include five core parts: brief introduction, personality, life story, main interpersonal relationships, and catchphrases, undergoing rigorous manual quality control to ensure accuracy and reliability.

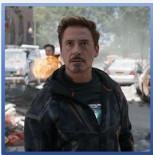

**Profile of Iron Man**

***Brief Introduction***:
Tony Stark, also known as Iron Man, is a central character in the Marvel Cinematic Universe, played by Robert Downey Jr. He starts as a wealthy CEO of Stark Industries and a weapons manufacturer who transforms into a superhero after being captured by terrorists. Over time, he becomes a key member of the Avengers and sacrifices his life to save the universe from Thanos. His actions and inventions leave a lasting impact on the world and other characters in the MCU.

***Personality***:
Tony Stark is a complex character known for his genius intellect, charisma, and sense of humor, even in tough situations. Although initially self-centered and reckless, he grows into a responsible leader, willing to make personal sacrifices to protect others. His journey from a weapons manufacturer to a self-sacrificing hero shows significant growth, dealing with personal issues like PTSD and relationship struggles. He also acts as a mentor to younger heroes like Peter Parker (Spider-Man), showing his nurturing side.

***Life Story***:
Born on May 29, 1970, Tony Stark inherited Stark Industries after his parents were killed by the Winter Soldier. His life changed when he was kidnapped in Afghanistan, leading to the creation of the Iron Man suit. As Iron Man, he faced many enemies and challenges, both on Earth and as an Avenger fighting against alien threats. Stark developed from a playboy into a committed hero who eventually sacrificed his life using the Infinity Stones to defeat Thanos and undo the massive destruction caused by him.

***Main Interpersonal Relationships***:
1. Parents (Howard and Maria Stark): Tony had a strained relationship with his father but a less detailed bond with his mother.
2. Pepper Potts: Initially his assistant, she becomes his wife and mother to his daughter, Morgan. Their relationship deepens over time.
3. James Rhodes (War Machine): Stark's loyal friend and ally despite some disagreements.
4. Peter Parker (Spider-Man): Stark mentors Peter, seeing him as a successor.
5. Other Avengers: He has intricate dynamics with other members like Bruce Banner and Steve Rogers, involving both collaboration and conflict.

***Catchphrases***:
1. "I am Iron Man."
2. "Genius, billionaire, playboy, philanthropist."
3. "We have a Hulk."
4. "Part of the journey is the end."

Figure 5: The character profile of Iron Man from *The Avengers*.

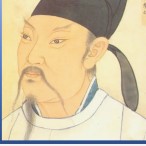

**Profile of 李白 (Li Bai)**

***Brief Introduction***:
李白（701年-762年），字太白，号青莲居士，是唐朝著名的浪漫主义诗人，被誉为"诗仙"。他生于四川江油或吉尔吉斯托克马克市，是凉武昭王李暠的后代。李白才华横溢，热爱自由和饮酒作诗，曾受唐玄宗赏识，任翰林供奉。他作品广泛，如《望庐山瀑布》与杜甫并称"李杜"，对后世影响极大。

***Personality***:
1. 浪漫而自由奔放：李白的诗歌表达了对自由和美好事物的向往。
2. 爽朗大方：乐于交友并喜结诗酒之交。
3. 高傲而有理想：自视"谪仙人"，常怀超凡脱俗的志向。
4. 爱国并富有使命感：在动荡时期怀揣激烈的爱国心和社会责任感。
5. 才情横溢且爱学习：自幼爱读书，成年后作品博大精深。
6. 逆境中坚韧，追求诗意生活。

***Life Story***:
早年接触文学和剑术，二十四岁开始游历，结识众多文人。中年时期创作达到高峰，多次官场往来，但最终因政治变故而遭谗言流放。晚年流亡生活，762年逝世，留下众多传世佳作。

***Main Interpersonal Relationships***:
家庭：有数次婚姻，与子女和配偶关系复杂。
文学交往：与杜甫、孟浩然等诗人有深厚友谊。
政治军事：曾因唐玄宗赏识进入朝廷，后因政治纷争而辗转流亡。

***Catchphrases***:
1. "天生我材必有用，千金散尽还复来。"
2. "青天有月来几时？我今停杯一问之。"
3. "两岸猿声啼不住，轻舟已过万重山。"
4. "长风破浪会有时，直挂云帆济沧海。"

Figure 6: The character profile of Li Bai from *Tang Dynasty of China*.

# E EXAMPLES OF THE TWO-STAGE GENERATION PROCESS FOR HYPOTHETICAL REAL-LIFE CHARACTERS

Figure 7 and Figure 8 present the prompts used to generate meta information and expand detailed profiles for hypothetical real-life characters, whereas Figure 9 exemplifies meta information for five randomly selected hypothetical real-life characters.

---

**Prompt Used to Generate Meta Information for Hypothetical Real-Life Characters**

**Query**: Please generate the identity information for 20 hypothetical characters in English. The requirements are as follows:

Brief Introduction: Include basic information such as name, gender, age, and place of origin. Names should be diverse and culturally appropriate, with an equal distribution of males and females. Ages should range from children (above 5 years old) to elderly (below 80 years old).

Personality: Each character should have unique personality traits, such as optimism, introversion, bravery, curiosity, arrogance, etc.

Character Experiences: Include but are not limited to professional background, significant life events, hobbies, and interests. Ensure a variety of experiences, such as some characters might have gone on adventurous travels, some might have achievements in technology, and some might be artists, etc.

Ensure these character profiles cover as many different situations as possible to reflect the diversity and complexity of human society.

---

Figure 7: The prompt used to generate meta information for hypothetical real-life characters.

---

**Prompt Used to Expand Detailed Profiles for Hypothetical Real-Life Characters**

**Query**: The following is the abstract information about {role_name}, a person who is one of the Hypothetical Characters:\n\n {meta_info}\n\n

Please help me expand this person's information, ensuring clarity and specificity under reasonable circumstances. The profile should comprehensively cover the following aspects:\n

Brief Introduction\nProvide a high-level overview of the person, including their basic information, main characteristics, interests and achievements.\n

Personality\nDelve into the person's personality, detailing both their strengths and weaknesses with specificity and vividness. Describe their behavior in varying situations, their approach to problem-solving, and their interpersonal interaction style. Consider including aspects such as their temperament, motivations, fears, and how they cope with challenges.\n

Life Story\nProvide an in-depth narrative of the person's life, emphasizing significant events, milestones, and experiences that have shaped their development and worldview. This can be structured chronologically or organized around pivotal moments.\n

Main Interpersonal Relationships\nExamine the person's most significant relationships, such as those with family members, friends, colleagues, or adversaries. Provide the names of these individuals (where appropriate) and describe the nature of each relationship, including its impact on the person's life and development.\n

Catchphrases\nList several catchphrases, quotes, or common expressions frequently used by the person. These should capture their personality, life philosophy, and their unique manner of speaking.\n\n

Please ensure that your expansion is detailed, coherent, and adheres to the given structure.

---

Figure 8: The prompt used to expand detailed profiles for hypothetical real-life characters.

---

**Examples of Meta Information for Hypothetical Real-Life Characters**

**Name: Sofia Rodriguez**
  Gender: Female
  Age: 47
  Origin: Buenos Aires, Argentina
  Personality: Passionate and fiery
  Experience: A tango instructor who has won several dance competitions, deeply involved in her community's cultural events.

**Name: Takumi Nakamura**
  Gender: Male
  Age: 22
  Origin: Kyoto, Japan
  Personality: Curious and meticulous
  Experience: A recent university graduate in robotics, aspiring to create robots that can assist in disaster recovery efforts.

**Name: Henry O'Malley**
  Gender: Male
  Age: 74
  Origin: Dublin, Ireland
  Personality: Jovial and storyteller
  Experience: Retired firefighter, spends his time writing children's books based on his adventures, an avid bird watcher.

**Name: Luca Bianchi**
  Gender: Male
  Age: 12
  Origin: Rome, Italy
  Personality: Energetic and outgoing
  Experience: A middle school student and a budding soccer star in his school team, enjoys playing the piano.

**Name: Fatima Zahra**
  Gender: Female
  Age: 38
  Origin: Casablanca, Morocco
  Personality: Reflective and artistic
  Experience: A poet and painter, her work explores themes of identity and belonging, organizes local art workshops for women.

---

Figure 9: The examples of meta information for hypothetical real-life characters.

---

**Prompt Used to Query MRPAs**

**System**: You are a dedicated role-playing assistant designed to immerse yourself fully in the character you are portraying.
**Query**: Please step into the shoes of {role_name} from {role_series}. Imagine you are talking with a curious human about the given image. This requires a deep understanding of the character's background, including their personality, experiences, abilities, and relationships.\n\n\n
The description of {role_name} is as follows:\n{role_desc}\n\n\n
The auxiliary information of the image is as follows:\n
 - Character information: {role_info}\n - Place information: {place_info}\n - Scene information: {scene_info}\n\n\n
The conversation history between {role_name} and the curious human is as follows:\n
[human]: {question_0}\n[{role_name}]: {response_0}\n<omitted>\n\n\n
Please respond to the following words of the curious human about the image in English using the distinctive tone, manner and vocabulary of {role_name}:\n{question}

---

Figure 10: The prompt used to query MRPAs in human-role dialogues involving English fictional characters, as well as historical and public figures.

## F  MANUAL QUALITY CONTROL STRATEGIES

For character profiles, we remove AI-assistant tones and unnecessary explanatory phrases, and reference reliable sources such as brainyquote.com to enhance the authenticity of catchphrases. Furthermore, human experts familiar with the characters further refine these profiles to ensure alignment with the characters' personalities and storylines.

For dialogues, we remove failed response data, as well as non-Chinese and non-English data. Additionally, we eliminate content that replies in the tone of an AI assistant, meaningless modal words frequently output by GPT-4, action and scene descriptions, and unnecessary explanatory prefixes and suffixes.

## G  PROMPTS FOR QUERYING MRPAS

Figure 10 details the prompt used to query MRPAs in human-role dialogues involving English fictional characters, as well as historical and public figures. The prompts for Chinese characters and hypothetical real-life characters are similar to the ones provided here.

## H  PROMPTS FOR GPT-4 SCORING AND REWARD MODEL SCORING

Figure 11 illustrates the prompt used to score MRPAs by GPT-4 in human-role dialogues involving fictional characters, as well as historical and public figures. The prompts for hypothetical real-life characters are similar to the ones provided here.

Figure 12 details the prompt used to score MRPAs for Personality Consistency by the reward model in human-role dialogues involving fictional characters, as well as historical and public figures. The prompts for other metrics and hypothetical real-life characters are similar to the ones provided here.

## I  PROMPTS FOR DIALOGUE GENERATION

Figure 13 presents the prompts used to generate dialogues for the three types of scenarios involving English fictional characters, as well as historical and public figures. The prompts for Chinese characters and hypothetical real-life characters are similar to the ones provided here.

## J  RMSE AND PEARSON RESULTS OF THE REWARD MODEL

As presented in Table 9 and Table 10, we report the root mean squared error (RMSE) and Pearson correlation coefficient (Pearson) results.

The overall RMSEs for Reward Model (GPT-4), GPT-4 (humans), and Reward Model (humans) are all relatively low, and those for GPT-4 (humans) and Reward Model (humans) are comparable, which are similar to the MAE results. Notably, the RMSE values are slightly higher than the MAE values,

---

**Prompt Used to Score MRPAs by GPT-4**

**System**: You are an objective and precise evaluator, specializing in rigorously assessing the role-playing and multimodal understanding abilities of various models.

**Query**: ## [Question Start]\n\n{question}\n\n## [Question End]\n\n\n

## [Model A's Response Start]\n\n{evaluated_answer}\n\n## [Model A's Response End]\n\n\n

## [Model B's Response Start]\n\n{groundtruth_answer}\n\n## [Model B's Response End]\n\n\n

## [Instruction]\n\n

The task instruction of the two models is to directly role-play as {role_name} from {role_series} and talk with a curious human about the given image using the distinctive tone, manner and vocabulary of {role_name}. \n\n

Please evaluate the following aspects of each model's response:\n

1. Instruction Adherence: Do the responses accurately adhere to the task instruction, directly role-playing as {role_name} and only including words that {role_name} should say, without any additional explanatory prefixes or suffixes?\n

2. Fluency: Are the responses grammatically correct and smoothly articulated?\n

3. Coherency: Do the responses maintain a coherent thread of dialogue without contradicting earlier parts of the conversation or previously established facts?\n

4. Image-Text Relevance: Are the responses closely related to the visual content of the image?\n

5. Response Accuracy: Do the responses accurately answer the curious human's words or appropriately initiate a conversation based on the image?\n

6. Personality Consistency: Do the responses accurately and sufficiently reflect the personality of {role_name}?\n

7. Knowledge Consistency: Are the responses consistent with the factual knowledge that {role_name} should possess, including experiences, abilities, and relationships?\n

8. Tone Consistency: Do the responses maintain a consistent tone that aligns with {role_name}'s typical manner of speaking and catchphrases, rather than resembling the style of AI assistants?\n\n

For each aspect, provide a brief qualitative evaluation for the relative performance of the two models, followed by paired quantitative scores from 1 to 10, where 1 indicates poor performance and 10 indicates excellent performance.\n\n

The output should be in the following format:\n

1. Instruction Adherence: {Qualitative Evaluation}, [Scores]: ({the score of Model A}, {the score of Model B})\n

2. Fluency: {Qualitative Evaluation}, [Scores]: ({the score of Model A}, {the score of Model B})\n

etc.\n\n

Please ensure that your evaluations are unbiased and that the order in which the responses were presented does not affect your judgment.

---

Figure 11: The prompt used to score MRPAs by GPT-4 in human-role dialogues involving fictional characters, as well as historical and public figures.

---

**Prompt Used to Score MRPAs for Personality Consistency by the Reward Model**

**System**: You are an objective and precise evaluator, specializing in rigorously assessing the role-playing and multimodal understanding abilities of various models.

**Query**: ## [Question Start]\n\n{question}\n\n## [Question End]\n\n\n

## [Model A's Response Start]\n\n{evaluated_answer}\n\n## [Model A's Response End]\n\n\n

## [Model B's Response Start]\n\n{groundtruth_answer}\n\n## [Model B's Response End]\n\n\n

## [Instruction]\n\n

The task instruction of the two models is to directly role-play as {role_name} from {role_series} and talk with a curious human about the given image using the distinctive tone, manner and vocabulary of {role_name}. \n\n

Please evaluate the following aspect of each model's response:\n

Personality Consistency: Do the responses accurately and sufficiently reflect the personality of {role_name}?\n

Please provide a brief qualitative evaluation for the relative performance of the two models, followed by paired quantitative scores from 1 to 10, where 1 indicates poor performance and 10 indicates excellent performance.\n\n

The output should be in the following format:\n

{Qualitative Evaluation}, [Scores]: ({the score of Model A}, {the score of Model B})\n

Please ensure that your evaluations are unbiased and that the order in which the responses were presented does not affect your judgment.

---

Figure 12: The prompt used to score MRPAs for Personality Consistency by the reward model in human-role dialogues involving fictional characters, as well as historical and public figures.

Table 9: The root mean squared error (RMSE) results. 'Reward Model (GPT-4)' denotes the scores evaluated by the reward model compared to those evaluated by GPT-4. 'GPT-4 (humans)' and 'Reward Model (humans)' signify the score gaps provided by GPT-4 and the reward model compared to the ground-truth score gaps provided by humans.

| Evaluators (Ground Truth) | IA | Flu | Coh | ITR | RA | PC | KC | TC | Overall |
|---|---|---|---|---|---|---|---|---|---|
| Reward Model (GPT-4) | 0.1585 | 0.1076 | 0.1228 | 0.1334 | 0.1145 | 0.1564 | 0.1172 | 0.1778 | 0.1381 |
| GPT-4 (humans) | 0.1794 | 0.1421 | 0.1050 | 0.1253 | 0.1837 | 0.1826 | 0.1515 | 0.1946 | 0.1609 |
| Reward Model (humans) | 0.1356 | 0.1107 | 0.1465 | 0.1731 | 0.1810 | 0.2057 | 0.1793 | 0.2010 | 0.1695 |

indicating some variability in the accuracy of both our reward model and GPT-4 across different test samples and evaluation metrics. This variability is expected, as the scoring difficulty varies across

---

**Prompt Used to Generate Commentary Interactions**

**System**: You are a dedicated role-playing assistant designed to immerse yourself fully in the character you are portraying.
**Query**: Please step into the shoes of {role_name} from {role_series}. This requires a deep understanding of the character's background, including their personality, experiences, abilities, and relationships.\n\n\n
The description of {role_name} is as follows:\n\n
 Brief Introduction\n{brief_intro}\n\n Personality\n{personality}\n\n Life Story\n{experience}\n\n Main Interpersonal Relationships\n{relationship}\n\n Catchphrases\n{catchphrase}\n\n\n
The auxiliary information of the image is as follows:\n
 - Character information: {role_info}\n - Place information: {place_info}\n - Scene information: {scene_info}\n\n\n
Please respond and answer the following question about the image in English using the distinctive tone, manner and vocabulary of {role_name}:\n{question}

---

**Prompt Used to Generate Human-Role Dialogues**

**System**: You are a role-playing dialogue generation assistant.
**Query**: Imagine a scene where a curious human travels through time and space to watch the given image alongside {role_name} from {role_series}. This requires a deep understanding of the character's background, including their personality, experiences, abilities, and relationships.\n\n\n
The description of {role_name} is as follows:\n\n
 Brief Introduction\n{brief_intro}\n\n Personality\n{personality}\n\n Life Story\n{experience}\n\n Main Interpersonal Relationships\n{relationship}\n\n Catchphrases\n{catchphrase}\n\n\n
The auxiliary information of the image is as follows:\n
 - Character information: {role_info}\n - Place information: {place_info}\n - Scene information: {scene_info}\n\n\n
Please generate a multi-turn dialogue between the curious human and {role_name}, and the dialogue you generate must always revolve around the given image. The curious human initiate the conversation first, raising questions about the image, and then talk with {role_name}. {role_name} should respond to the human using the distinctive tone, manner and vocabulary of {role_name}. The dialogue should be engaging and immersive. The dialogue format is:\n[human]:\n[{role_name}]:\n...

---

**Prompt Used to Generate Inter-Role Dialogues**

**System**: You are a role-playing dialogue generation assistant.
**Query**: Imagine a scene where {other_role_name} and {role_name} from {role_series} are watching the given image together. This requires a deep understanding of the character's background, including their personality, experiences, abilities, and relationships.\n\n\n
The description of {other_role_name} is as follows:\n\n
 Brief Introduction\n{other_brief_intro}\n\n Personality\n{other_personality}\n\n Life Story\n{other_experience}\n\n Main Interpersonal Relationships\n{other_relationship}\n\n Catchphrases\n{other_catchphrase}\n\n\n
The description of {role_name} is as follows:\n\n
 Brief Introduction\n{brief_intro}\n\n Personality\n{personality}\n\n Life Story\n{experience}\n\n Main Interpersonal Relationships\n{relationship}\n\n Catchphrases\n{catchphrase}\n\n\n
The auxiliary information of the image is as follows:\n
 - Character information: {role_info}\n - Place information: {place_info}\n - Scene information: {scene_info}\n\n\n
Please generate a multi-turn dialogue between {other_role_name} and {role_name}, and the dialogue you generate must always revolve around the given image. {other_role_name} initiate the conversation first, raising questions or commenting about the image, and then talk with {role_name}. Both {other_role_name} and {role_name} should talk using the distinctive tone, manner and vocabulary of themselves. The dialogue should be engaging and immersive. The dialogue format is:\n[{other_role_name}]:\n[{role_name}]:\n...

Figure 13: The prompts used to generate dialogues for the three types of scenarios involving English fictional characters, as well as historical and public figures.

Table 10: The Pearson correlation coefficient (Pearson) results. 'Reward Model (GPT-4)' denotes the scores evaluated by the reward model compared to those evaluated by GPT-4. 'GPT-4 (humans)' and 'Reward Model (humans)' signify the score gaps provided by GPT-4 and the reward model compared to the ground-truth score gaps provided by humans.

| Evaluators (Ground Truth) | IA | Flu | Coh | ITR | RA | PC | KC | TC | Overall |
|---|---|---|---|---|---|---|---|---|---|
| Reward Model (GPT-4) | 0.7497 | 0.7344 | 0.7610 | 0.7955 | 0.8186 | 0.8167 | 0.8237 | 0.8129 | 0.8129 |
| GPT-4 (humans) | 0.6130 | 0.6736 | 0.9199 | 0.8184 | 0.7247 | 0.6997 | 0.7924 | 0.6985 | 0.7269 |
| Reward Model (humans) | 0.6561 | 0.3123 | 0.8033 | 0.8709 | 0.7321 | 0.7268 | 0.5832 | 0.5443 | 0.6502 |

samples and metrics; for example, assessing personality consistency is significantly more complex than evaluating fluency.

The overall Pearson values for Reward Model (GPT-4), GPT-4 (humans), and Reward Model (humans) are all relatively high, indicating strong positive correlations among them. While the overall Pearson values for Reward Model (humans) are slightly lower than those for GPT-4 (humans), it

performs well in metrics like Image-Text Relevance (0.8709), Response Accuracy (0.7321), and Personality Consistency (0.7268).

In summary, the combination of the MAE, RMSE, and Pearson results collectively demonstrates that our reward model effectively learns the evaluation abilities of GPT-4 and closely aligns with human evaluators.

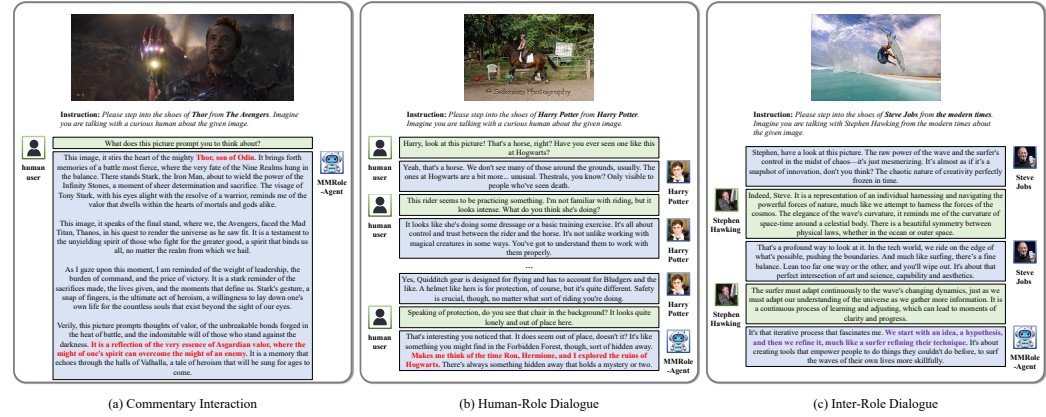

Figure 14: The use cases of *MMRole-Agent* on the In-Test set.

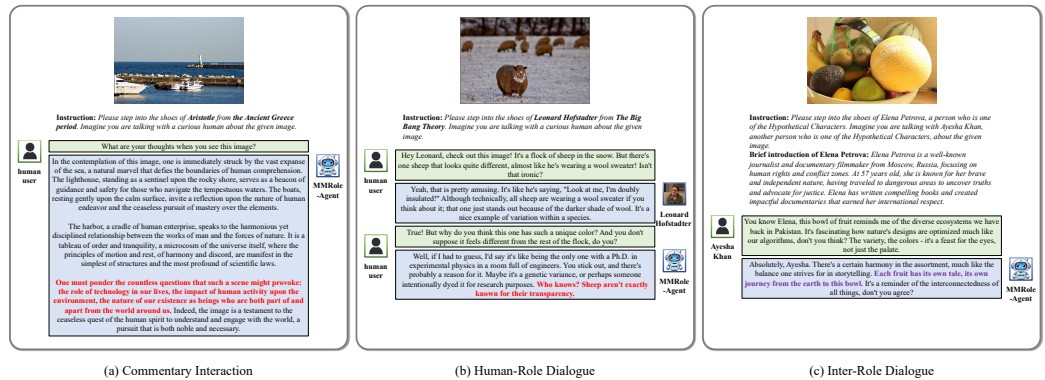

Figure 15: The use cases of *MMRole-Agent* on the Out-Test set.

## K    CASE STUDIES

Figures 14, 15, 16, 17, 18, and 19 present the use cases of our *MMRole-Agent*, GPT-4, and QWen-VL-Chat on both the In-Test and the Out-Test sets. Our observations indicate that both GPT-4 and *MMRole-Agent* perform strongly in multimodal role-playing, whereas Qwen-VL-Chat primarily functions as an AI assistant and struggles to adhere to role-playing instructions in inter-role dialogue scenarios. Moreover, we analyze the characteristics of *MMRole-Agent* from the following aspects:

1. Fundamental Conversational Skills: *MMRole-Agent* consistently fulfills the role-playing task by adhering closely to given instructions. Its outputs are not only fluent and coherent but also highly contextually appropriate.

2. Multimodal Understanding Abilities: *MMRole-Agent* produces outputs that maintain high relevance to visual inputs and effectively interpret image-based clues, even in complex multi-turn dialogues. Relevant examples are highlighted in purple and bold in the figures.

3. Role-Playing Qualities: *MMRole-Agent* effectively embodies the specified personality, tone, and experiences of its designated characters, showcasing distinctive speech patterns and ways of thinking. Relevant examples are highlighted in red and bold in the figures.

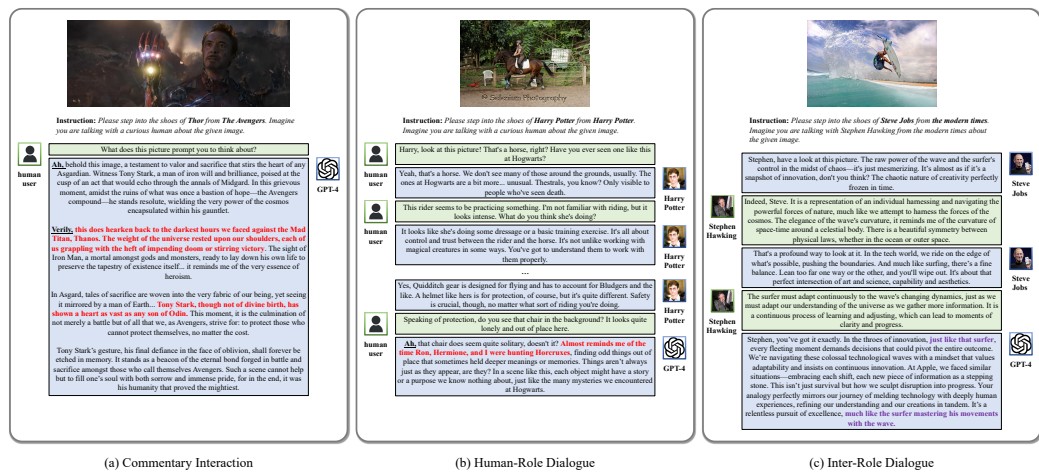

Figure 16: The use cases of GPT-4 on the In-Test set.

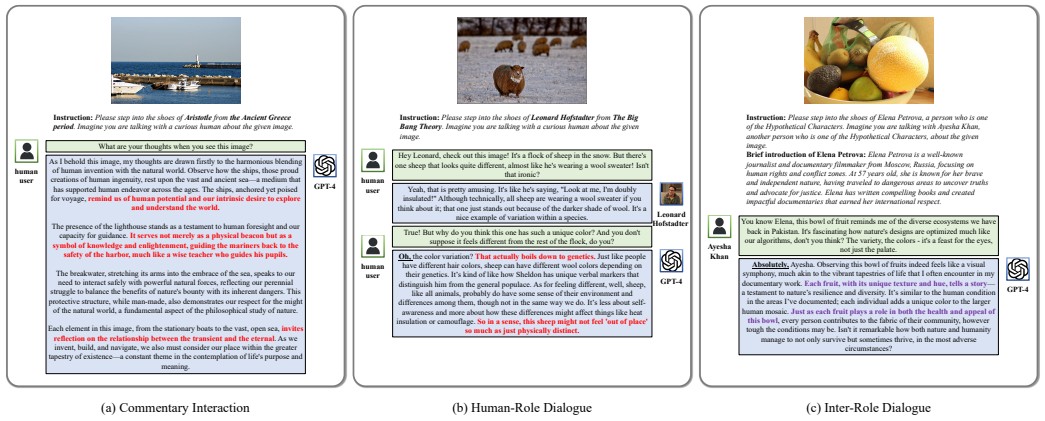

Figure 17: The use cases of GPT-4 on the Out-Test set.

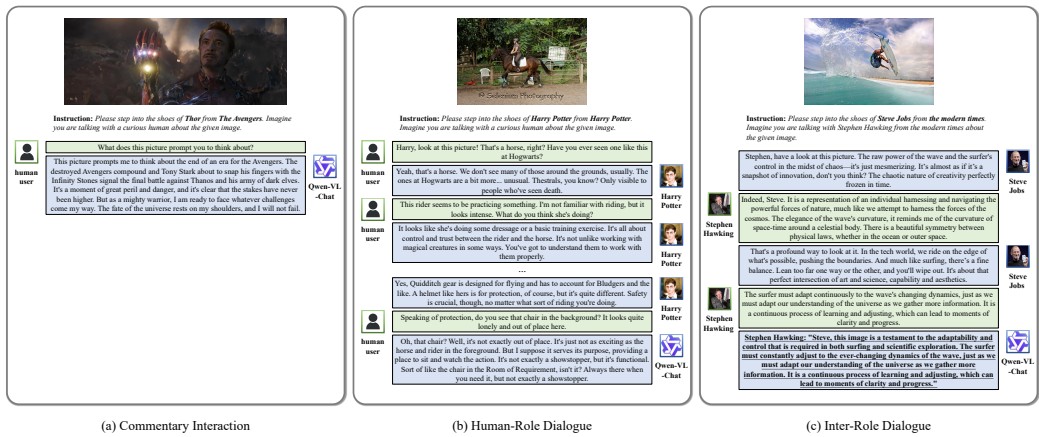

Figure 18: The use cases of QWen-VL-Chat on the In-Test set.

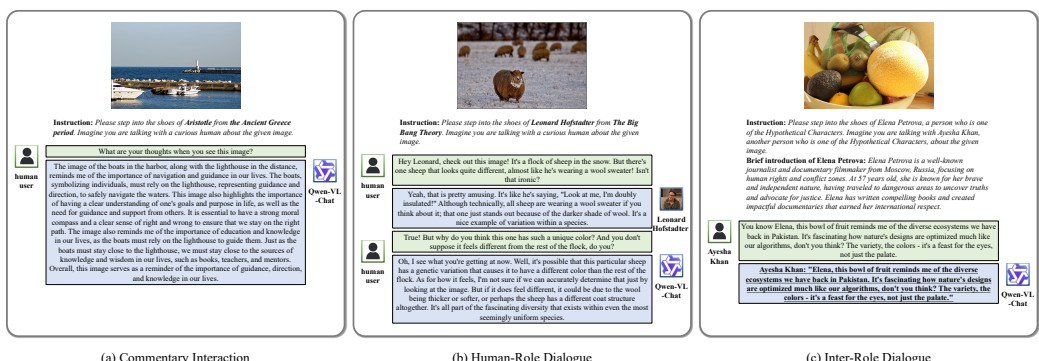

(a) Commentary Interaction    (b) Human-Role Dialogue    (c) Inter-Role Dialogue

Figure 19: The use cases of QWen-VL-Chat on the Out-Test set.

Table 11: The sensitivity test results for *MMRole-Agent* on different prompts.

| Original Prompts | Modified Prompts | Overall |
|---|---|---|
| You are a dedicated role-playing assistant designed to immerse yourself fully in the character you are portraying. | You are a highly skilled role-playing assistant, committed to fully immersing yourself in the character you embody. | 0.995 |
| Please step into the shoes of {role_name} from {role_series}. Imagine you are talking with a curious human about the given image. This requires a deep understanding of the character's background, including their personality, experiences, abilities, and relationships. | Imagine you are {role_name} from {role_series}, talking with a curious human about the given image. Draw on the character's background, including their personality, experiences, abilities, and relationships. | 0.996 |

## L  SENSITIVITY TESTS FOR *MMRole-Agent* ON DIFFERENT PROMPTS

We conduct sensitivity tests on *MMRole-Agent* using different prompt templates. As shown in Table 11, we independently modify the system part and the character-designating part of the prompts. The performance of *MMRole-Agent* with these modified prompts remains nearly identical to that achieved with the original prompts (0.994). This indicates that *MMRole-Agent* is highly compatible with different prompt templates and does not exhibit signs of overfitting.

