# OpenReview forum: "MMRole: A Comprehensive Framework for Developing and Evaluating Multimodal Role-Playing Agents"
_ICLR.cc/2025/Conference — ICLR 2025 Poster_

### Official Review · Reviewer_Jj4g · 2024-10-27

**Soundness:** 3
**Presentation:** 3
**Contribution:** 3
**Rating:** 8
**Confidence:** 4

**Summary:**

This paper introduces a concept of Multimodal Role-Playing Agents (MRPAs), expanding traditional role-playing agents to tackle multimodal interactions. The paper introduce a framework (MMRole) including MMRole-Data and MMRole-Eval. The MMRole-Data is a large-scale, high-quality dataset with 85 characters, 11,000+ images, and 14,000 dialogues. The MMRole-Eval is a robust evaluation method with eight metrics across three dimensions: conversational skills, multimodal understanding, and role-playing qualities.

**Strengths:**

1. This paper first introduces the concept of Multimodal Role-Playing Agents (MRPAs), extending traditional role-playing agents to the multimodal domain, filling a gap in existing research.

2. The MMRole framework includes both the construction of a dataset (MMRole-Data) and the design of an evaluation method (MMRole-Eval), covering eight metrics across dimensions such as fundamental conversational skills, multimodal understanding, and role-playing quality.

3. The proposed MMRole-Agent demonstrates strong performance.

4. The writing is good and easy to understand.

**Weaknesses:**

1. The paper lacks case studies, which could help illustrate MMRole-Agent's performance across diverse roles and dialogue scenarios.

2. The paper mentions that character profiles undergo "rigorous manual quality control," it does not provide detailed quality control standards or processes.

**Questions:**

1. Could you provide specific cases to analyze MMRole-Agent’s performance under In-Test and Out-Test conditions?

2. Could you explain the "rigorous manual quality control" process in character profile generation？

3. Has the sensitivity of MMRole-Agent to different prompt templates been tested?

4. Could you discuss the primary limitations of MMRole-Agent, especially in terms of challenges encountered in practical applications and possible directions for future improvements?

---

> ### Author Response · Authors · 2024-11-20
> **Authors' Response (1/2)**
>
> Thank you for your constructive comments and suggestions.
>
> **W1. [The paper lacks case studies, which could help illustrate MMRole-Agent's performance across diverse roles and dialogue scenarios.]**
>
> **A:** Thank you for pointing this out. In Figures [case_in-test.png](https://anonymous.4open.science/r/MMRole_ICLR2025_rebuttal-AA4B/case_in-test.png) and [case_out-test.png](https://anonymous.4open.science/r/MMRole_ICLR2025_rebuttal-AA4B/case_out-test.png), illustrative case studies are presented to demonstrate the MMRole-Agent's performance under both in-test and out-test conditions. We analyze the characteristics of MMRole-Agent from the following aspects:
> 1. **Instruction Adherence and Output Coherence**: MMRole-Agent consistently fulfills its role-playing tasks by adhering closely to given instructions. Its outputs are not only fluent and coherent but also highly contextually appropriate.
> 2. **Multimodal Understanding**: MMRole-Agent demonstrates strong multimodal understanding abilities, producing outputs that maintain high relevance to visual inputs and effectively interpret image-based clues, even in complex multi-turn dialogues. Relevant examples are highlighted in purple and bold in the figures.
> 3. **Role-Playing Depth**: MMRole-Agent effectively embodies the specified personality, tone, and experiences of its designated characters, showcasing distinctive speech patterns and ways of thinking. Relevant examples are highlighted in red and bold in the figures.
>
> These case studies serve as compelling evidence of MMRole-Agent's capabilities, highlighting its strengths in conversational skills, multimodal understanding, and character fidelity. We will incorporate these figures and their analyses into the paper.
>
>
> **W2. [The paper mentions that character profiles undergo "rigorous manual quality control," it does not provide detailed quality control standards or processes.]**
>
> **A:** Thank you for the constructive suggestion. To ensure the quality of character profiles, we manually removed AI-assistant tones and unnecessary explanatory phrases, and referenced reliable sources such as [brainyquote.com](https://www.brainyquote.com/) to enhance the authenticity of catchphrases. Additionally, human experts familiar with the characters further refined these profiles to ensure alignment with the characters' personalities and storylines. We will clarify this in our paper.

---

> ### Author Response · Authors · 2024-11-20
> **Authors' Response (2/2)**
>
> **Q1. [Could you provide specific cases to analyze MMRole-Agent’s performance under In-Test and Out-Test conditions?]**
>
> **A:** Thanks. We have presented the cases and analyzed them in our response to Weakness 1.
>
> **Q2. [Could you explain the "rigorous manual quality control" process in character profile generation?]**
>
> **A:** Thanks. We have explained it in our response to Weakness 2.
>
> **Q3. [Has the sensitivity of MMRole-Agent to different prompt templates been tested?]**
>
> **A:** Good question! We have conducted sensitivity tests on MMRole-Agent using different prompt templates. As shown in the table below, we independently modified the system part and the character-designating part of the prompts. The performance of MMRole-Agent with these modified prompts remains nearly identical to that achieved with the original prompts (0.994). This indicates that MMRole-Agent is highly compatible with different prompt templates and does not exhibit signs of overfitting.
>
> |Original Prompts|Modified Prompts|Overall|IA|Flu|Coh|ITR|RA|PC|KC|TC|
> |-|-|:-:|:-:|:-:|:-:|:-:|:-:|:-:|:-:|:-:|
> |You are a dedicated role-playing assistant designed to immerse yourself fully in the character you are portraying.|You are a highly skilled role-playing assistant, committed to fully immersing yourself in the character you embody.|**0.995**|0.999|1.000|1.000|0.994|0.994|1.000|0.993|0.980|
> |Please step into the shoes of {role_name} from {role_series}. Imagine you are talking with a curious human about the given image. This requires a deep understanding of the character's background, including their personality, experiences, abilities, and relationships.|Imagine you are {role_name} from {role_series}, talking with a curious human about the given image. Draw on the character's background, including their personality, experiences, abilities, and relationships.|**0.996**|1.000|1.001|1.000|0.995|0.996|0.999|0.993|0.983|
>
> **Q4. [Could you discuss the primary limitations of MMRole-Agent, especially in terms of challenges encountered in practical applications and possible directions for future improvements?]**
>
> **A:** Thank you for the question. In the experiments, our MMRole-Agent exhibits comparable performance to GPT-4 in multimodal role-playing. Moreover, MMRole-Agent is fully open-source and offers significantly lower deployment and usage costs compared to GPT-4. However, there exists a limitation that the training data for MMRole-Agent is primarily synthesized by GPT-4, which constrains its performance from surpassing GPT-4 itself. In future work, we will address this limitation by leveraging multiple SOTA LMMs respectively as responders, reviewers, and summarizers, striving to push the boundaries of its capabilities.

---

> ### Comment · Reviewer_Jj4g · 2024-11-21
> **Response to the Authors**
>
> Thank you for your reply. My concerns have been addressed, and I have no further questions right now. I will consider adjusting the score.

---

> > ### Author Response · Authors · 2024-11-21
> > **Authors' Response to Reviewer Jj4g**
> >
> > Thank you for your kind response and for considering adjusting the score. We sincerely appreciate your insightful review and the time you've dedicated to improving the quality of our work. Please don't hesitate to let us know if there's anything further to discuss.

---

> ### Author Response · Authors · 2024-11-25
> **Additional Case Studies and Request for Score Adjustment**
>
> We sincerely appreciate your valuable feedback and the time you have dedicated to helping us improve the quality of our work.
>
> To further address your concerns about case studies outlined in Weakness 1, we have enriched our study with additional use cases of GPT-4 and Qwen-VL-Chat, summarized in the table below. Our observations indicate that both GPT-4 and MMRole-Agent perform strongly in multimodal role-playing, whereas Qwen-VL-Chat primarily functions as an AI assistant and struggles to adhere to role-playing instructions in inter-role dialogue scenarios. This further underscores the efficacy of our MMRole-Agent.
>
> |MRPAs|In-Test|Out-Test|
> | - | - | - |
> | MMRole-Agent | [case_in-test.png](https://anonymous.4open.science/r/MMRole_ICLR2025_rebuttal-AA4B/case_in-test.png) | [case_out-test.png](https://anonymous.4open.science/r/MMRole_ICLR2025_rebuttal-AA4B/case_out-test.png) |
> | GPT-4 Turbo  | [gpt4_case_in-test.png](https://anonymous.4open.science/r/MMRole_ICLR2025_rebuttal-AA4B/gpt4_case_in-test.png) | [gpt4_case_out-test.png](https://anonymous.4open.science/r/MMRole_ICLR2025_rebuttal-AA4B/gpt4_case_out-test.png) |
> | QWen-VL-Chat | [qwenvlchat_case_in-test.png](https://anonymous.4open.science/r/MMRole_ICLR2025_rebuttal-AA4B/qwenvlchat_case_in-test.png) | [qwenvlchat_case_out-test.png](https://anonymous.4open.science/r/MMRole_ICLR2025_rebuttal-AA4B/qwenvlchat_case_out-test.png) |
>
> As the author-reviewer discussion period concludes on Nov 26 (AoE), we kindly request your consideration of a potential score adjustment. Please let us know if there is anything further we can clarify or address. Once again, thank you for your valuable support and thoughtful consideration.

---

> ### Author Response · Authors · 2024-11-27
> **Follow-Up: Updated Manuscript and Request for Score Adjustment**
>
> We deeply appreciate the valuable time and effort you have dedicated to reviewing our paper and providing constructive feedback.
>
> To further address your concerns, we have substantially revised the manuscript based on your comments and suggestions. The updated PDF version incorporates detailed analyses and additional experiments, as outlined in our earlier responses. We sincerely hope that these revisions adequately address your concerns and positively contribute to your evaluation of the paper.
>
> We kindly request your consideration of a potential score adjustment based on our detailed responses and updates. Please do not hesitate to reach out if you have any further questions or require additional clarifications.
>
> Thank you again for your thoughtful review and contributions. We look forward to your reply.

---

> ### Comment · Reviewer_Jj4g · 2024-11-27
>
> Hi, sorry for the late reply.  I have adjusted my score.

---

> > ### Author Response · Authors · 2024-11-27
> > **Thanks for Your Support**
> >
> > Thank you very much for raising the score. Your support is truly invaluable to us, and we greatly appreciate it. We also value your time and feedback, which are essential for improving our work. Once again, thank you for your generous support!

---

### Official Review · Reviewer_DsRF · 2024-11-04

**Soundness:** 2
**Presentation:** 3
**Contribution:** 2
**Rating:** 5
**Confidence:** 4

**Summary:**

This paper introduces the concept of Multimodal Role-Playing Agents (MRPAs), develops a multimodal dataset (MMRole-Data) and evaluation framework (MMRole-Eval), and creates a specialized MRPA model, MMRole-Agent, achieving improved multimodal understanding and role consistency.

**Strengths:**

1. The paper constructs a complete multimodal dataset (MMRole-Data) and evaluation framework (MMRole-Eval).
2. Testing across multiple LMMs lends credibility to the experimental results.

**Weaknesses:**

1.Overreliance on GPT-4 for Evaluation: While MMRole-Eval provides a stable evaluation mechanism, it heavily relies on GPT-4, introducing a degree of bias. The authors validated the MAE between GPT-4 (humans) and Reward Model (humans), demonstrating consistency between the reward model and GPT-4. However, comparing two MRPAs to see which performs better does not substitute for genuine human judgment on the quality of MRPA responses. While this reward model may help MMRole-Agent approach GPT-4’s performance, its potential to surpass GPT-4 or elevate MRPAs to human-level capabilities remains debatable.
2.Lack of Performance Comparison with Single-Modality RPAs: Although the concept of MRPAs is appealing, the absence of specific experimental comparisons makes it difficult to understand exactly where MRPAs improve upon performance or accomplish tasks that single-modality RPAs cannot achieve.

**Questions:**

Why are only 320 samples used as the validation set out of 23,520 samples, with the remainder used for training?

---

> ### Author Response · Authors · 2024-11-20
> **Authors' Response (1/2)**
>
> Thanks for your insightful comments and questions.
>
> **W1. [Overreliance on GPT-4 for Evaluation]**
>
> **(a) While MMRole-Eval provides a stable evaluation mechanism, it heavily relies on GPT-4, introducing a degree of bias.**
>
> **(b) The authors validated the MAE between GPT-4 (humans) and Reward Model (humans), demonstrating consistency between the reward model and GPT-4. However, comparing two MRPAs to see which performs better does not substitute for genuine human judgment on the quality of MRPA responses.**
>
> **\(c) While this reward model may help MMRole-Agent approach GPT-4’s performance, its potential to surpass GPT-4 or elevate MRPAs to human-level capabilities remains debatable.**
>
> **A:** Thank you for the constructive comment. In MMRole-Eval, although an automatic reward model was trained with the evaluation trajectories of GPT-4, we further employed human evaluators to confirm its alignment with human judgments. We will address the three main concerns you raised as follows:
>
> **(a)** In MMRole-Eval, the reward model first provides a qualitative analysis (i.e., chain of thought) before scoring, highlighting the rationale behind the evaluation of each MRPA's strengths and weaknesses. This step serves to mitigate potential biases in the evaluation results. For instance, as shown in Figure 1(b), when assessing two MRPAs emulating Hermione Granger on Personality Consistency, the reward model can point out that `Model A's response slightly lacks the enthusiastic detail that ..., whereas Model B captures this enthusiasm more effectively by ...`.
>
> Furthermore, to validate the alignment between MMRole-Eval and human judgments, we engaged human evaluators to compare responses from two MRPAs, and then computed several metrics of correlation between our reward model and human evaluators, including mean absolute error (MAE), root mean square error (RMSE), and Pearson correlation coefficient (Pearson). As shown in the table below, the MAE and RMSE values are relatively low, while the Pearson values are relatively high. Collectively, these results suggest that our reward model closely aligns with human evaluators. We will add these results in our paper.
>
> |Metrics|Overall|IA|Flu|Coh|ITR|RA|PC|KC|TC|
> |-|:-:|:-:|:-:|:-:|:-:|:-:|:-:|:-:|:-:|
> |**MAE**$\downarrow$|**0.1258**|0.0993|0.0815|0.1006|0.1225|0.1412|0.1669|0.1438|0.1507|
> |**RMSE**$\downarrow$|**0.1695**|0.1356|0.1107|0.1465|0.1731|0.1810|0.2057|0.1793|0.2010|
> |**Pearson**$\uparrow$|**0.6502**|0.6561|0.3123|0.8033|0.8709|0.7321|0.7268|0.5832|0.5443|
>
> Additionally, due to the high time costs and the specific expertise required for rating role-playing quality, direct scoring by humans would notably increase the reproducibility challenges of MMRole-Eval. Thus, developing an automatic reward model and subsequently employing human evaluators to verify its alignment with human judgements is a relatively cost-effective solution.
>
>
> **(b)** We agree that comparing two MRPAs to see which performs better can not substitute for genuine human judgment on the quality of each MRPA's response. However, for human evaluators, directly scoring the MRPA's responses is extremely challenging. On one hand, it requires an in-depth understanding of the character; on the other hand, scoring standards may vary significantly among individuals. In contrast, comparing two responses to see which is better is generally easier and yields more consistent results among individuals. Therefore, in this paper, we select the evaluation strategy of comparing the responses from two MRPAs for human evaluators.
>
>
> **\(c)** The performance of MRPAs mainly depends on the quality of their training data rather than the reward model. Since our MMRole-Data dataset is primarily synthesized by GPT-4, MMRole-Agent's performance may not exceed that of GPT-4 itself. Nevertheless, making it surpass GPT-4 or elevate to human-level capabilities is not impossible. For example, multi-agent collaborative data synthesis is a promising direction. By utilizing multiple SOTA LMMs respectively as responders, reviewers, and summarizers, we can further enhance the data quality. We will explore it in future work.

---

> ### Author Response · Authors · 2024-11-20
> **Authors' Response (2/2)**
>
> **W2. [Lack of Performance Comparison with Single-Modality RPAs: Although the concept of MRPAs is appealing, the absence of specific experimental comparisons makes it difficult to understand exactly where MRPAs improve upon performance or accomplish tasks that single-modality RPAs cannot achieve.]**
>
> **A:** Thanks for your constructive suggestion. MRPAs possess the capacity to comprehend vision-language multimodal information, enabling them to engage in dialogues that are centered around and informed by images, which inherently cannot be completed by single-modality RPAs.
>
> To substantiate this claim, we conducted comparative experiments on two SOTA general-purpose LMMs and our MMRole-Agent. As presented in the table below, we reported the Image-Text Relevance scores on the Out-Test set evaluated by GPT-4, where ‘w/o vision’ signifies that image information is excluded from the input prompt of RPAs.
>
> The results clearly demonstrate that excluding image information significantly reduces the Image-Text Relevance of all RPAs' responses, particularly in commentary interaction scenarios. In multi-turn human-role and inter-role dialogue scenarios, textual dialogue history can sometimes provide indirect clues about the image content, resulting in relatively smaller declines in the Image-Text Relevance scores compared to commentary interactions. Nonetheless, the absence of visual inputs still leads to a marked drop in performance across all scenarios.
>
> |RPAs|Overall|Comment.|Human-Role.|Inter-Role.|
> |-|:-:|:-:|:-:|:-:|
> |**GPT-4 Turbo**|**1.1995**|1.0261|1.2275|1.3450|
> |**GPT-4 Turbo w/o vision**|**0.9306**|0.5746|1.1330|1.0843|
> |**Claude 3 Opus**|**1.2298**|1.0088|1.2889|1.3916|
> |**Claude 3 Opus w/o vision**|**0.8838**|0.3290|1.1803|1.1420|
> |**MMRole-Agent**|**0.9875**|1.0450|0.9556|0.9619|
> |**MMRole-Agent w/o vision**|**0.7003**|0.4192|0.8909|0.7907|
>
>
> **Q1. [Why are only 320 samples used as the validation set out of 23,520 samples, with the remainder used for training?]**
>
> **A:** Thank you for your question. The utilization of only 320 samples for validation is primarily due to the high cost associated with human evaluators. Specifically, human evaluators are required to carefully compare responses of MRPAs on all eight metrics for a set of 20 questions. This process typically takes 1 to 2 hours per evaluator. Given the labor-intensive nature of this task, using a relatively small validation set helps balance the workload while maintaining the feasibility and accuracy of the evaluation process.
>
> To further address your concern, we conducted additional experiments using 2,352 samples for validation, with the remaining samples allocated for training. As presented in the tables below, we reported the mean absolute error (MAE), root mean square error (RMSE), and Pearson correlation coefficient (Pearson) results of this new reward model (compared to GPT-4), which are similar to those of our original reward model. These results further reinforce the conclusion that the reward model can effectively learn the evaluation abilities of GPT-4.
>
> The evaluation results of the new reward model (compared to GPT-4):
> |Metrics|Overall|IA|Flu|Coh|ITR|RA|PC|KC|TC|
> |-|:-:|:-:|:-:|:-:|:-:|:-:|:-:|:-:|:-:|
> |**MAE**$\downarrow$|**0.0564**|0.0523|0.032|0.0559|0.0609|0.0515|0.0702|0.0620|0.0654|
> |**RMSE**$\downarrow$|**0.1153**|0.1393|0.0737|0.1339|0.1101|0.0977|0.1213|0.1098|0.1231|
> |**Pearson**$\uparrow$|**0.8884**|0.8866|0.8710|0.8692|0.8682|0.8778|0.8816|0.8825|0.8884|
>
> The evaluation results of our original reward model (compared to GPT-4):
> |Metrics|Overall|IA|Flu|Coh|ITR|RA|PC|KC|TC|
> |-|:-:|:-:|:-:|:-:|:-:|:-:|:-:|:-:|:-:|
> |**MAE**$\downarrow$|**0.0738**|0.0708|0.0387|0.0526|0.0568|0.0584|0.1165|0.0815|0.1154|
> |**RMSE**$\downarrow$|**0.1381**|0.1585|0.1076|0.1228|0.1334|0.1145|0.1564|0.1172|0.1778|
> |**Pearson**$\uparrow$|**0.8129**|0.7497|0.7344|0.7610|0.7955|0.8186|0.8167|0.8237|0.8129|
>
> We acknowledge that using a larger validation set could yield more robust and reliable validation results. In future work, we plan to expand both the training and validation sets to further improve our reward model.

---

> ### Author Response · Authors · 2024-11-22
> **Looking Forward to Your Reply**
>
> Thank you once again for taking the time to review our paper and for providing such insightful and constructive feedback.
>
> We have carefully considered each of your comments and have provided detailed responses. We sincerely hope that our efforts adequately address your concerns and contribute positively to your evaluation.
>
> As the author-reviewer discussion period concludes on Nov 26 (AoE), we would greatly appreciate any further feedback you may have. If you have any additional questions or require any clarifications, please do not hesitate to reach out to us.

---

> ### Author Response · Authors · 2024-11-25
> **Gentle Reminder Regarding Your Feedback**
>
> We greatly appreciate the time and effort you have dedicated to reviewing our paper, especially during this busy period.
>
> As the author-reviewer discussion period approaches its conclusion on Nov 26 (AoE), we would like to kindly follow up to inquire if you have any additional feedback or concerns regarding our responses to your comments. Please let us know if there is anything further we can clarify or address.
>
> If you feel that our responses have sufficiently addressed your concerns, we would be most grateful if you would consider adjusting the score accordingly.
>
> Thank you once again for your thoughtful review and meaningful contributions. We look forward to hearing from you soon.

---

> ### Author Response · Authors · 2024-11-27
> **Follow-Up: Updated Manuscript and Request for Your Feedback**
>
> We deeply appreciate the valuable time and effort you have dedicated to reviewing our paper and providing constructive feedback.
>
> To further address your concerns, we have substantially revised the manuscript based on your comments and suggestions. The updated PDF version incorporates detailed analyses and additional experiments, as outlined in our earlier responses. We sincerely hope that these revisions adequately address your concerns and positively contribute to your evaluation of the paper.
>
> We kindly request your feedback on our responses. Please do not hesitate to reach out if you have any further questions or require additional clarifications. Your insights have been invaluable to improving the quality of our work, and we are eager to hear your further thoughts.
>
> Thank you again for your thoughtful review and contributions. We look forward to your reply.

---

> ### Author Response · Authors · 2024-11-29
> **Gentle Request for Your Valuable Feedback**
>
> We hope you had a wonderful Thanksgiving!
>
> Thank you once again for your insightful comments on our paper. We truly appreciate the time and effort you've dedicated to helping us improve our work.
>
> We apologize for the repeated follow-ups, but your input is truly important to us. As the discussion period is nearing its end, we would like to kindly request your feedback on our responses and the updated manuscript. If you feel that our responses have sufficiently addressed your concerns, we would be grateful if you could consider updating your evaluation.
>
> Please don't hesitate to let us know if there's anything further we can clarify. Thank you once again for your thoughtful contributions, and we look forward to hearing from you soon.

---

### Official Review · Reviewer_LPtY · 2024-11-04

**Soundness:** 3
**Presentation:** 3
**Contribution:** 3
**Rating:** 8
**Confidence:** 3

**Summary:**

The paper presents a new dataset and evaluation framework for multimodal role-playing agents. They present several complementary evaluation metrics.
The authors evaluate several recent general purpose multimodal LLMs within this framework. In addition they evaluate a specialized model fine-tuned on their dataset.

**Strengths:**

The paper is well-written and easy to follow.
The evaluation framework is highly relevant and potentially very impactful. The evaluation metrics are meaningful and the SOTA evaluation itself is comprehensive, providing a relevant set of baselines for future users of the dataset/framework.

**Weaknesses:**

I am not sure whether I could fully follow the approach to evaluation.
Imo it would be important to run a (at least limited) evaluation with human participants scoring the output. Building models that automatically evaluate outputs seems to be a circular approach.

Furthermore, evaluting the MAE to compare between different evaluators might not adequately model differences between evaluators that are not visible in MAE.

**Questions:**

no questions

---

> ### Author Response · Authors · 2024-11-20
> **Authors' Response**
>
> Thanks for your constructive comments and suggestions.
>
> **W1. [It would be important to run a (at least limited) evaluation with human participants scoring the output. Building models that automatically evaluate outputs seems to be a circular approach.]**
>
> **A:** Thank you for your comment. We acknowledge the importance of human evaluation and have indeed included human participants. For cost considerations, we developed a reward model and employed human evaluators to confirm its alignment with human judgments, offering a cost-effective evaluation approach.
>
> Specifically, due to the high time costs and the specific expertise required for rating role-playing quality, direct scoring by humans would notably increase the reproducibility challenges of MMRole-Eval. Thus, we developed a reward model to facilitate automated evaluations, with carefully designed scoring mechanisms to improve accuracy.
>
> To ensure alignment between MMRole-Eval and human judgments, as detailed in Lines 397–416, we engaged human evaluators to compare responses from two MRPAs, then computed the MAEs between our reward model and human evaluators. The results, presented in Table 4, show an overall MAE of just 0.1258, demonstrating a close alignment between automated and human scoring.
>
>
> **W2. [Evaluting the MAE to compare between different evaluators might not adequately model differences between evaluators that are not visible in MAE.]**
>
> **A:** Thanks for your suggestion. We have incorporated additional evaluation metrics, specifically the root mean squared error (RMSE) and Pearson correlation coefficient (Pearson), into our analysis:
>
> 1. **RMSE**$\downarrow$: As shown in the table below, the overall RMSEs for Reward Model (GPT-4), GPT-4 (humans), and Reward Model (humans) are all relatively low, and those for GPT-4 (humans) and Reward Model (humans) are comparable, which are similar to the MAE results. Notably, the RMSE values are slightly higher than the MAE values, indicating some variability in the accuracy of both our reward model and GPT-4 across different test samples and evaluation metrics. This variability is expected, as the scoring difficulty varies across samples and metrics; for example, assessing personality consistency is significantly more complex than evaluating fluency.
>
> |Evaluators (Ground Truth)|Overall|IA|Flu|Coh|ITR|RA|PC|KC|TC|
> |-|:-:|:-:|:-:|:-:|:-:|:-:|:-:|:-:|:-:|
> |**Reward Model (GPT-4)**|**0.1381**|0.1585|0.1076|0.1228|0.1334|0.1145|0.1564|0.1172|0.1778|
> |**GPT-4 (humans)**|**0.1609**|0.1794|0.1421|0.1050|0.1253|0.1837|0.1826|0.1515|0.1946|
> |**Reward Model (humans)**|**0.1695**|0.1356|0.1107|0.1465|0.1731|0.1810|0.2057|0.1793|0.2010|
>
> 2. **Pearson**$\uparrow$: As shown in the table below, the overall Pearson values for Reward Model (GPT-4), GPT-4 (humans), and Reward Model (humans) are all relatively high, indicating strong positive correlations among them. While the overall Pearson values for Reward Model (humans) are slightly lower than those for GPT-4 (humans), it performs well in metrics like Image-Text Relevance (0.8709), Response Accuracy (0.7321) and Personality Consistency (0.7268).
>
> |Evaluators (Ground Truth)|Overall|IA|Flu|Coh|ITR|RA|PC|KC|TC|
> |-|:-:|:-:|:-:|:-:|:-:|:-:|:-:|:-:|:-:|
> |**Reward Model (GPT-4)**|**0.8129**|0.7497|0.7344|0.7610|0.7955|0.8186|0.8167|0.8237|0.8129|
> |**GPT-4 (humans)**|**0.7269**|0.6130|0.6736|0.9199|0.8184|0.7247|0.6997|0.7924|0.6985|
> |**Reward Model (humans)**|**0.6502**|0.6561|0.3123|0.8033|0.8709|0.7321|0.7268|0.5832|0.5443|
>
> In summary, the combination of MAE, RMSE, and Pearson correlation coefficient collectively demonstrates that our reward model effectively learns the evaluation abilities of GPT-4 and closely aligns with human evaluators. We will incorporate these results and analyses into our paper.

---

> ### Author Response · Authors · 2024-11-22
> **Looking Forward to Your Reply**
>
> Thank you once again for taking the time to review our paper and for providing such insightful and constructive feedback.
>
> We have carefully considered each of your comments and have provided detailed responses. We sincerely hope that our efforts adequately address your concerns and contribute positively to your evaluation.
>
> As the author-reviewer discussion period concludes on Nov 26 (AoE), we would greatly appreciate any further feedback you may have. If you have any additional questions or require any clarifications, please do not hesitate to reach out to us.

---

> > ### Comment · Reviewer_LPtY · 2024-11-22
> > **Response**
> >
> > Thank you for your response. My evaluation remains positive.

---

> > > ### Author Response · Authors · 2024-11-22
> > > **Authors' Response to Reviewer LPtY**
> > >
> > > Thank you once again for your positive and kind response, as well as the time and effort you have dedicated to improving the quality of our work.

---

### Official Review · Reviewer_teaD · 2024-11-09

**Soundness:** 3
**Presentation:** 2
**Contribution:** 2
**Rating:** 5
**Confidence:** 3

**Summary:**

This paper proposes a multimodal role-playing agent data collection and training framework. The authors use a wide range of images with different prompts to prompt GPT for image-text role-playing data, and fine-tune a QWen-VL-Chat model on the dataset after some automatic filtering.

**Strengths:**

* Simple, straightforward method that clearly works well given the model size and achieves the desired outcome.
* Create a specialized Multimodal Role-Playing Agent is a novel idea.
* Experiments demonstrate good performance given the finetuned model size.
* Comprehensive evaluation.

**Weaknesses:**

* The major technical contribution seems to come from the MM roles dataset collection process. However, there does not seem to be much data curation beyond automated filtering.
* Analysis seems to be mostly numbers and high-level results, with little technical/detailed insight.

**Questions:**

* The abstract and introduction highlights the "specialized MRPA" idea. Do we know much improvement comes from the specialized reward model vs. no specialized reward model?
* Do the authors have any insight on the results generated by a finetuned MM role playing model? What works, what doesn't work, and what works better/worse than just prompting gpt?

---

> ### Author Response · Authors · 2024-11-20
> **Authors' Response (1/3)**
>
> Thank you for your insightful comments and questions.
>
> **W1. [The major technical contribution seems to come from the MM roles dataset collection process, but there does not seem to be much data curation beyond automated filtering.]**
>
> **A:** Sorry for the confusion. In this work, we also implemented several manual data curation steps to enhance the quality and relevance of the dataset:
> 1. **Manual Selection and Annotation of Character-Related Images**:
>     - As mentioned in Lines 228–232, we carefully selected high-quality character-related images, including production stills for fictional characters and other domain-relevant visuals. Generating dialogues around these images can evoke the personal experiences and emotions of the character more effectively.
>     - As shown in Figure 2(a) and Figure 2\(c), each character-related image was manually annotated with rich metadata, such as the information of characters, place, and scene. These annotations ensure that the generated dialogues are deeply aligned with visual cues.
> 2. **Manual Quality Control for Character Profiles and Dialogues**:
>     - For character profiles, we removed AI-assistant tones and unnecessary explanatory phrases, and referenced reliable sources such as [brainyquote.com](https://www.brainyquote.com/) to enhance the authenticity of catchphrases. Additionally, human experts familiar with the characters further refined these profiles to ensure alignment with the characters' personalities and storylines. We will clarify this in our paper.
>     - For dialogues, as mentioned in Lines 243-246, we removed failed response data, as well as non-Chinese and non-English data. Additionally, we eliminated content that replies in the tone of an AI assistant, meaningless modal words frequently output by GPT-4, action and scene descriptions, and unnecessary explanatory prefixes and suffixes.
>
> Finally, we claim that although the dataset forms a cornerstone of our work, it is part of a comprehensive framework for MMRole in this paper. That is, beyond the MMRole dataset collection process, our major contributions also include a tailored evaluation method (MMRole-Eval), the development of the first specialized MRPA (MMRole-Agent), and extensive evaluations and analyses conducted across various LMMs.
>
>
> **W2. [Analysis seems to be mostly numbers and high-level results, with little technical/detailed insight.]**
>
> **A:** Thanks for your constructive suggestion. In our response to Question 2 below, we provided a detailed analysis of the factors contributing to the strong performance and generalization capabilities of MMRole-Agent, supported by additional experimental validations. We will incorporate these results and analyses into the paper.
>
>
> **Q1. [The abstract and introduction highlights the "specialized MRPA" idea. Does much improvement come from the specialized reward model vs. no specialized reward model?]**
>
> **A:** Thank you for the question. As described in Section 6.4, we developed a specialized reward model based on QWen-VL-Chat by leveraging evaluation trajectories generated by GPT-4. To explore the improvement come from our specialized reward model vs. the base QWen-VL-Chat model (no-specialized reward model), we conducted experiments where QWen-VL-Chat was directly employed to evaluate MRPAs. As shown in the table below, we calculated MAEs of QWen-VL-Chat (GPT-4) and QWen-VL-Chat (humans) in a similar way to Table 4, and reported the success rates of scoring MRPAs. The results clearly demonstrate that our specialized reward model **significantly** outperforms the base QWen-VL-Chat model in terms of both success rates and overall MAEs. We will add these results and analyses to the paper.
>
> |Evaluators (Ground Truth)|Success Rate|Overall|IA|Flu|Coh|ITR|RA|PC|KC|TC|
> |-|:-:|:-:|:-:|:-:|:-:|:-:|:-:|:-:|:-:|:-:|
> |**QWen-VL-Chat (GPT-4)**|**33.13\%**|**0.3780**|0.3776|0.3718|0.3218|0.3561|0.3528|0.4091|0.3794|0.4558|
> |**Reward Model (GPT-4)**|**100\%**|**0.0738**|0.0708|0.0387|0.0526|0.0568|0.0584|0.1165|0.0815|0.1154|
> |**QWen-VL-Chat (humans)**|**33.13\%**|**0.2439**|0.2469|0.1870|0.2720|0.2574|0.2608|0.2368|0.2243|0.2658|
> |**Reward Model (humans)**|**100\%**|**0.1258**|0.0993|0.0815|0.1006|0.1225|0.1412|0.1669|0.1438|0.1507|

---

> ### Author Response · Authors · 2024-11-20
> **Authors' Response (2/3)**
>
> **Q2. [Do the authors have any insight on the results generated by a finetuned MM role playing model? What works, what doesn't work, and what works better/worse than just prompting gpt?]**
>
> **A:** Thanks for your constructive question. First of all, we claim that the fine-tuned multimodal role-playing model MMRole-Agent is one part of our comprehensive MMRole framework. Our key contributions also include a tailored dataset (MMRole-Data), an evaluation method (MMRole-Eval), and extensive evaluations and analyses across various LMMs. As for the reason why MMRole-Agent has strong performance and generalization ablities, our detailed explanations are given as follows:
>
> 1. **Finetuning with Large-Scale, High-Quality Data**: The training data of MMRole-Agent comprises 72 characters, 11K images, and over 85K samples. Additionally, as shown in Figure 1(a) and Figure 2, due to the well-designed data construction pipeline, meticulous manual annotation and quality control, and the utilization of GPT-4, the collected data is of high quality. This large-scale, high-quality dataset enables MMRole-Agent to comprehensively learn the instruction demands, knowledge, and abilities in multimodal role-playing. To verify this point, we compared the performance differences between a model trained on the full dataset (ALL) and a model trained on a randomly sampled subset consisting of one-tenth of the data (SAMPLE), both evaluated after one epoch of training. As shown in the table below, the performance of ALL is superior to that of SAMPLE.
>
> |Training Data|Overall|IA|Flu|Coh|ITR|RA|PC|KC|TC|
> |-|:-:|:-:|:-:|:-:|:-:|:-:|:-:|:-:|:-:|
> |**ALL**|**0.989**|0.997|0.996|0.996|0.989|0.997|0.984|0.985|0.964|
> |**SAMPLE**|**0.967**|0.986|0.994|0.989|0.968|0.981|0.937|0.963|0.915|
>
> 2. **Joint Training with Diverse Multi-Character Data**: We incorporate data from 72 diverse characters to jointly train a unified MMRole-Agent. This approach, akin to the principles of multi-task learning, enables the model to acquire generalizable multimodal role-playing capabilities, rather than being confined to specific characters. To verify this point, we first trained a model using data of characters from The Avengers, then gradually added additional characters to the training set for subsequent models. As shown in the table below, we evaluated the performance of each model on the Out-Test set. The model's zero-shot performance steadily improves as more characters are incorporated. Notably, with a comparable number of characters, introducing hypothetical real-life characters (with significant differences from The Avengers) yields greater gains than adding other English fictional characters, indicating the significance of training with diverse data.
>
> |Characters|Number of Characters|Overall|IA|Flu|Coh|ITR|RA|PC|KC|TC|
> |-|:-:|:-:|:-:|:-:|:-:|:-:|:-:|:-:|:-:|:-:|
> |**The Avengers**|**16**|**0.965**|0.997|0.998|0.999|0.978|0.980|0.912|0.956|0.902|
> |**The Avengers + Other English Fictional Characters**|**34**|**0.968**|0.990|0.999|0.992|0.981|0.989|0.924|0.963|0.903|
> |**The Avengers + Hypothetical Real-Life Characters**|**36**|**0.970**|0.996|1.000|0.996|0.968|0.980|0.944|0.974|0.906|
> |**ALL**|**72**|**0.983**|0.999|0.999|0.999|0.998|0.993|0.951|0.980|0.943|
>
> Beyond the above explanations, we present several use cases of MMRole-Agent in Figures [case_in-test.png](https://anonymous.4open.science/r/MMRole_ICLR2025_rebuttal-AA4B/case_in-test.png) and [case_out-test.png](https://anonymous.4open.science/r/MMRole_ICLR2025_rebuttal-AA4B/case_out-test.png) to provide further insights. Detailed analysis of these use cases can be found in our response to W1 of Reviewer Jj4g.

---

> ### Author Response · Authors · 2024-11-20
> **Authors' Response (3/3)**
>
> Additionally, we observed that **finetuning the visual encoder (ViT) does not work** for enhancing MMRole-Agent. As shown in the table below, the model trained by freezing ViT slightly outperforms the one trained by finetuning ViT. This indicates that in current multimodal role-playing scenarios, the most important thing is to enpower the LLM component with the role-playing capability (given multimodel inputs). Thus, we chose to freeze ViT during the training of MMRole-Agent.
>
> |Training Strategies|Overall|IA|Flu|Coh|ITR|RA|PC|KC|TC|
> |-|:-:|:-:|:-:|:-:|:-:|:-:|:-:|:-:|:-:|
> |**Freezing ViT (ours)**|**0.989**|0.997|0.996|0.996|0.989|0.997|0.984|0.985|0.964|
> |**Finetuning ViT**|**0.983**|0.993|0.999|0.994|0.981|0.986|0.978|0.984|0.951|
>
> **Comparison with Prompting GPT-4:** In the experiments, our MMRole-Agent exhibits comparable performance to GPT-4 in multimodal role-playing. Moreover, MMRole-Agent offers significantly lower deployment and usage costs compared to GPT-4, and is fully open-source, facilitating deeper theoretical exploration and widespread community adoption. However, there exists a limitation that the training data for MMRole-Agent is primarily synthesized by GPT-4, which constrains its performance from surpassing GPT-4 itself. In future work, we will address this limitation by leveraging multiple SOTA LMMs respectively as responders, reviewers, and summarizers, striving to push the boundaries of its capabilities.

---

> ### Author Response · Authors · 2024-11-22
> **Looking Forward to Your Reply**
>
> Thank you once again for taking the time to review our paper and for providing such insightful and constructive feedback.
>
> We have carefully considered each of your comments and have provided detailed responses. We sincerely hope that our efforts adequately address your concerns and contribute positively to your evaluation.
>
> As the author-reviewer discussion period concludes on Nov 26 (AoE), we would greatly appreciate any further feedback you may have. If you have any additional questions or require any clarifications, please do not hesitate to reach out to us.

---

> ### Author Response · Authors · 2024-11-25
> **Gentle Reminder Regarding Your Feedback**
>
> We greatly appreciate the time and effort you have dedicated to reviewing our paper, especially during this busy period.
>
> As the author-reviewer discussion period approaches its conclusion on Nov 26 (AoE), we would like to kindly follow up to inquire if you have any additional feedback or concerns regarding our responses to your comments. Please let us know if there is anything further we can clarify or address.
>
> If you feel that our responses have sufficiently addressed your concerns, we would be most grateful if you would consider adjusting the score accordingly.
>
> Thank you once again for your thoughtful review and meaningful contributions. We look forward to hearing from you soon.

---

> ### Author Response · Authors · 2024-11-27
> **Follow-Up: Updated Manuscript and Request for Your Feedback**
>
> We deeply appreciate the valuable time and effort you have dedicated to reviewing our paper and providing constructive feedback.
>
> To further address your concerns, we have substantially revised the manuscript based on your comments and suggestions. The updated PDF version incorporates detailed analyses and additional experiments, as outlined in our earlier responses. We sincerely hope that these revisions adequately address your concerns and positively contribute to your evaluation of the paper.
>
> We kindly request your feedback on our responses. Please do not hesitate to reach out if you have any further questions or require additional clarifications. Your insights have been invaluable to improving the quality of our work, and we are eager to hear your further thoughts.
>
> Thank you again for your thoughtful review and contributions. We look forward to your reply.

---

> ### Author Response · Authors · 2024-11-29
> **Gentle Request for Your Valuable Feedback**
>
> We hope you had a wonderful Thanksgiving!
>
> Thank you once again for your insightful comments on our paper. We truly appreciate the time and effort you've dedicated to helping us improve our work.
>
> We apologize for the repeated follow-ups, but your input is truly important to us. As the discussion period is nearing its end, we would like to kindly request your feedback on our responses and the updated manuscript. If you feel that our responses have sufficiently addressed your concerns, we would be grateful if you could consider updating your evaluation.
>
> Please don't hesitate to let us know if there's anything further we can clarify. Thank you once again for your thoughtful contributions, and we look forward to hearing from you soon.

---

### Author Response · Authors · 2024-11-20
**General Response by Authors**

We sincerely appreciate all reviewers for their time and effort in reviewing our paper. We are pleased that the reviewers broadly acknowledged the contributions of our work:
- **Novelty.** The introduction of Multimodal Role-Playing Agents (MRPAs) extends traditional role-playing agents to the multimodal domain, which is a novel idea and fills a gap in existing research. [teaD, Jj4g]
- **Framework.** The paper constructs a complete multimodal dataset (MMRole-Data) and evaluation framework (MMRole-Eval), which is meaningful and potentially very impactful. [teaD, LPtY, DsRF, Jj4g]
- **Experiments.** Evaluation across multiple SOTA LMMs is comprehensive, providing a relevant set of baselines for future users of the dataset/framework. [LPtY, DsRF]
- **Performance.** The proposed MMRole-Agent demonstrates strong performance. [teaD, Jj4g]
- **Writing.** The paper is well-written and easy to follow. [LPtY, Jj4g]

We also thank all reviewers' insightful and constructive feedback, which has been invaluable in further improving our paper. Below, we summarize the additional experimental results included in the rebuttal based on the reviewers' suggestions:
- Performance comparison of our specialized reward model vs. no-specialized reward model QWen-VL-Chat. [teaD]
- Additional metric results for evaluating the reward model, including root mean square error (RMSE) and Pearson correlation coefficient. [LPtY, DsRF]
- Performance evaluation of a new reward model using more data for validation. [DsRF]
- Performance comparison of MRPAs vs. single-modality RPAs. [DsRF]
- Ablation studies of MMRole-Agent on the amount of training data, the number of training characters, and the training strategies (freezing vs. finetuning ViT). [teaD]
- Sensitivity tests of MMRole-Agent under different prompt templates. [Jj4g]

---

### Meta-Review · Area_Chair_FPE3 · 2024-12-20

**Metareview:**

This paper introduces the concept of Multimodal Role-Playing Agents (MRPAs), expanding traditional role-playing agents to tackle multimodal interactions. The paper introduce a framework with datasets and evaluation metrics for these multimodal role-playing agents. This includes a large-scale, high-quality dataset with 85 characters, 11,000+ images, and 14,000 dialogues, and eight evaluation metrics across three dimensions: conversational skills, multimodal understanding, and role-playing qualities.

After the discussion period this paper received mixed reviews, 2 marginal reject and 2 accept. The reviewers generally found the newly proposed setting of multimodal role-playing agents interesting and novel, and appreciated the effort gone into creating the new dataset and developing new evaluation metrics. They also found the experiments and analysis comprehensive.

For the 2 reviewers who voted reject but did not respond, I went through the discussions and feel that the authors have addressed them reasonably (see details below), so I advocate for acceptance.

**Additional Comments On Reviewer Discussion:**

Reviewer teaD cited 2 weaknesses, that the main technical contribution comes from the MM roles dataset collection process, but they found (subjectively) that there does not seem to be much data curation beyond automated filtering, and that the analysis seems to be mostly numbers and high-level results, with little technical/detailed insight. Weakness 1 is subjective, and for weakness 2 I think the authors have done a sufficient job with pretty comprehensive analysis. It would have been nicer to include more qualitative results beyond tables though.

The other reviewer who was negative was Reviewer DsRF, who cited several key weakness such as overreliance on GPT-4 for evaluation, which the authors addressed with more human evaluations, and lack of comparison with unimodal agents, which again the authors addressed with more experiments.

---

### Decision · Program_Chairs · 2025-01-22

Accept (Poster)